# A suppression hierarchy among competing motor programs drives sequential grooming in *Drosophila*

Andrew M Seeds*, Primoz Ravbar, Phuong Chung, Stefanie Hampel, Frank M Midgley Jr†, Brett D Mensh, Julie H Simpson*

Janelia Farm Research Campus, Howard Hughes Medical Institute, Ashburn, United States

**Abstract** Motor sequences are formed through the serial execution of different movements, but how nervous systems implement this process remains largely unknown. We determined the organizational principles governing how dirty fruit flies groom their bodies with sequential movements. Using genetically targeted activation of neural subsets, we drove distinct motor programs that clean individual body parts. This enabled competition experiments revealing that the motor programs are organized into a suppression hierarchy; motor programs that occur first suppress those that occur later. Cleaning one body part reduces the sensory drive to its motor program, which relieves suppression of the next movement, allowing the grooming sequence to progress down the hierarchy. A model featuring independently evoked cleaning movements activated in parallel, but selected serially through hierarchical suppression, was successful in reproducing the grooming sequence. This provides the first example of an innate motor sequence implemented by the prevailing model for generating human action sequences.

*For correspondence: seeds. andrew@gmail.com (AMS); simpsonj@janelia.hhmi.org (JHS)

†Deceased

**Competing interests:** The authors declare that no competing interests exist.

**Reviewing editor**: Ronald L Calabrese, Emory University, United States

## Introduction

Animals engage in a variety of complex serial behaviors that are essential for survival, such as nest building, communication, courtship, and prey capture. Serial behaviors can be subdivided into distinct motor actions that are mutually exclusive and often proceed in a given sequence (*Gallistel, 1980*; *Adams, 1984*). While our understanding of how motor actions are produced expands (*Marder et al., 2005*; *Bizzi and Cheung, 2008*; *Grillner and Jessell, 2009*; *Costa, 2011*), less is known about the rules that govern how nervous systems assemble individual actions into coordinated sequential behaviors.

There are two main hypotheses about mechanisms that drive the sequential progression of actions during serial behavior. One hypothesis suggests that each action in a series triggers performance of the next action (*James, 1890*; *Adams, 1984*). This is one of the earliest views of motor sequencing and is thought to underlie some sequential behaviors, such as bird song (*Long et al., 2010*). An alternative hypothesis proposes that the premotor units of all actions in a sequence are activated or readied in parallel and then sequentially selected through winner-take-all competition (*Lashley, 1951*; *Houghton and Hartley, 1995*; *Bullock, 2004*). This model was originally proposed based on human psychology experiments demonstrating a parallel representation of letter typing acts prior to the typing of a word. Electrophysiological evidence in primates is consistent with this parallel model of serial behavior (*Averbeck et al., 2002*; *Mushiake et al., 2006*). However, it has been difficult to causatively confirm the model and determine its underlying action selection mechanisms.

One strategy for testing how serial behavior is executed is to first identify the individual motor actions that make up the behavior and to then use neural control over each action to investigate how its manipulation affects the progression of the other actions in the series. Independent control of each

**eLife digest** Anyone who has ever lived with a cat is familiar with its grooming behavior. This innate behavior follows a particular sequence as the cat methodically cleans its body parts one-by-one. Many animals also have grooming habits, even insects such as fruit flies. The fact that grooming sequences are seen across such different species suggests that this behavior is important for survival. Nevertheless, how the brain organizes grooming sequences, or other behaviors that involve a sequence of tasks, is not well understood.

Fruit flies make a good model for studying grooming behavior for a couple of reasons. First, they are fastidious cleaners. When coated with dust they will faithfully carry out a series of cleaning tasks to clean each body part. Second, there are many genetic tools and techniques that researchers can use to manipulate the fruit flies' behaviors. One technique allows specific brain cells to be targeted and activated to trigger particular behaviors.

Seeds et al. used these sophisticated techniques, computer modeling, and behavioral observations to uncover how the brains of fruit flies orchestrate a grooming sequence. Dust-covered flies follow a predictable sequence of cleaning tasks: beginning by using their front legs to clean their eyes, they then clean their antennae and head. This likely helps to protect their sensory organs. Next, they move on to the abdomen, possibly to ensure that dust doesn't interfere with their ability to breathe. Wings and thorax follow last. Periodically, the flies stop to rub their legs together to remove any accumulated dust before resuming the cleaning sequence.

Seeds et al. activated different sets of brain cells one-by-one to see if they could trigger a particular grooming task and found that individual cleaning tasks could be triggered, in the absence of dust, by stimulating a specific group of brain cells. This suggests each cleaning task is a discrete behavior controlled by a subset of cells. Then Seeds et al. tried to stimulate more than one cleaning behavior at a time; they discovered that wing-cleaning suppressed thorax-cleaning, abdomen-cleaning suppressed both of these, and head-cleaning suppressed all the others. This suggests that a 'hierarchy' exists in the brain that exactly matches the sequence that flies normally follow as they clean their body parts.

By learning more about how the brain coordinates grooming sequences, the findings of Seeds et al. may also provide insights into other behaviors that involve a sequence of tasks, such as nest building in animals or typing in humans. Following on from this work, one of the next challenges will be to see if such behaviors also use a 'suppression hierarchy' to ensure that individual tasks are carried out in the right order.

action in a sequence is challenging to achieve in most systems. However, recent tool development in *Drosophila melanogaster* permits subdivisions of behavior to be identified and manipulated in freely moving flies by expressing neural activators in subsets of neurons in the brain. This approach has revealed that acute neural activation can trigger specific behaviors and subcomponents of behaviors, such as abdominal bending and courtship song (two subcomponents of the courtship sequence) (*Clyne and Miesenböck, 2008*; *von Philipsborn et al., 2011*; *Flood et al., 2013*). While these tools have invigorated interest in identifying the neurons that drive behavior, they also enable experiments to probe the organization of serial behavior through specific control of individual motor actions.

Here, we study the organizational principles underlying the neural control of serial action selection in an innate behavior common to most limbed animals—grooming (*Sachs, 1988*). Grooming consists of discrete cleaning movements that occur in predictable sequences; however, the mechanisms that govern transitions between the movements are not known (*Szebenyi, 1969*; *Fentress and Stilwell, 1973*; *Dawkins and Dawkins, 1976*; *Sachs, 1988*). Local mechanical stimulation of a body part causes fruit flies to perform precisely targeted cleaning sweeps with their front or hind legs towards that body part (*Vandervorst and Ghysen, 1980*; *Corfas and Dudai, 1989*). In contrast, when they are completely coated in dust, flies coordinate a repertoire of cleaning movements to groom their whole bodies (*Figure 1A*; *Bentley, 1975*; *Phillis et al., 1993*). We reasoned that we could exploit the behavior of such dust-coated flies to determine how different grooming actions are prioritized and ordered when they are stimulated at the same time. We show that dust on the body elicits different independently evoked cleaning motor programs for five body parts that progress in series. By genetically

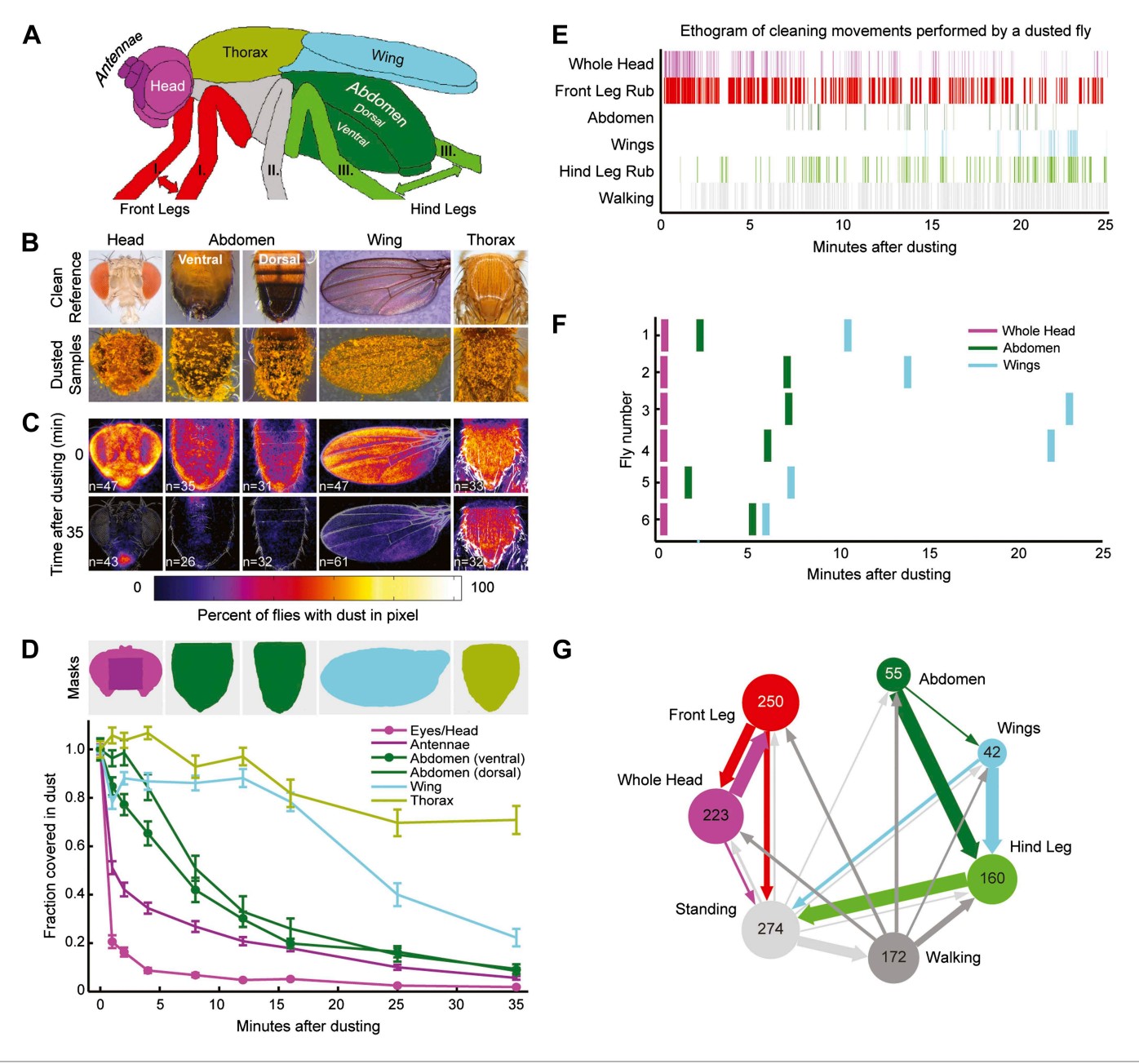

**Figure 1**. Wild-type flies clean different areas of the body sequentially. (**A**) Diagram of body parts cleaned by front leg (red hues) or hind leg (green hues) movements. (**B–D**) Dust distribution measurements of the bodies of flies that were coated in yellow dust and allowed to groom. (**B**) Body parts were imaged (dusted samples) and aligned to clean reference images in order to determine the fraction of dust left on each part. (**C**) Average spatial distribution of dust 0 min after dusting and after flies groomed for 35 min. The number of flies contributing to each heat map is displayed. (**D**) Dust removal across a 35-min time course. Masks define regions for counting the yellow pixels (dust) remaining on each sample. Each time point (normalized to 0-min samples) is plotted as the fraction of dust left in the defined regions and shown as the mean ± SEM; n ≥ 26 flies. Figure panel is compiled from data shown in *Figure 1—figure supplement 3*. (**E**) Representative ethogram of the five most common cleaning movements performed by an individual fly after dusting (manually scored from video recordings). All head cleaning movements are binned because eye and antennal cleaning are not easily distinguishable in the dusted state using our analysis methods (labeled whole head). (**F**) Latency to the first bout of head, abdomen, or wing cleaning after dusting for each of six flies annotated. (**G**) Transitions among different body cleaning movements, standing, and walking (across a 25-min time course, n = 6 flies). The radii of the nodes are proportional to the log of the average fraction of total cleaning bouts for each movement. Average total bouts for each movement are shown. Arrow widths represent the transition probabilities between the movements (displaying transition probabilities ≥0.05).

*Figure 1. Continued*

The following figure supplements are available for figure 1:

**Figure supplement 1**. Grooming apparatus for dusting, recording, and observing flies.

**Figure supplement 2**. Strategies for quantifying dust on the body surface.

**Figure supplement 3**. Wild-type flies remove dust from body parts at different rates.

**Figure supplement 4**. Sequential cleaning of the head, abdomen, and wings requires dust.

**Figure supplement 5**. Transitions among cleaning movements of dusted wild-type flies.

**Figure supplement 6**. Transitions among cleaning movements performed by dusted wild-type flies over a time course.

targeting different neural subsets, we gain experimental control over each cleaning motor program to demonstrate that they are organized into a suppression hierarchy. We next generate an empirically derived computational model that demonstrates how a suppression hierarchy can mediate the sequential selection of the cleaning motor programs. This work reveals how the organization of *Drosophila* grooming is well described by a parallel model of serial behavior.

## Results

### Dust-induced grooming is a sequential behavior

We tested whether flies would prioritize cleaning of specific body parts when they were completely dirty. This was achieved by coating flies in dust, paired with measurements of changing dust distributions on the body surface (*Figure 1B,C*, *Figure 1—figure supplements 1 and 2*) and manual annotation of their cleaning movements (*Figure 1E*; *Video 1*). In response to dust on their bodies, flies groomed with a predictable progression of targeted cleaning movements (*Figure 1D–F*, *Figure 1—figure supplement 3*, *Figure 1—figure supplement 4B,C*). This progression required dust, as undusted flies did not groom in a sequential progression (*Figure 1—figure supplement 4A,C*). Dusted flies rapidly cleaned their eyes and then focused on the antennae, demonstrating that they sequentially clean specific regions on the head (*Figure 1D*). They next cleaned their posterior body parts in the order from abdomen to wings to thorax. The grooming progression occurred as a sequence in which the probabilities of cleaning different regions of the body gradually changed through time. That is, the progression was not absolutely unidirectional as flies returned to cleaning earlier body parts even though they had already started cleaning later body parts (referred to as return cleaning) (*Figure 1E*, *Figure 1—figure supplement 4B*, example ethograms). As the flies progressed through the grooming sequence, each bout of body part cleaning featured cyclic transitions between sweeps of the targeted region and rubbing of the legs against each other (*Figure 1G*, *Figure 1—figure supplement 5*; *Video 1*). These cyclic bouts of leg rubbing likely occurred when the legs accumulated sufficient dust from sweeping the body, which was then rubbed towards the distal leg parts and removed. Grooming of the entire body did not necessarily proceed as a continuous process. Flies often paused to walk around and then resumed

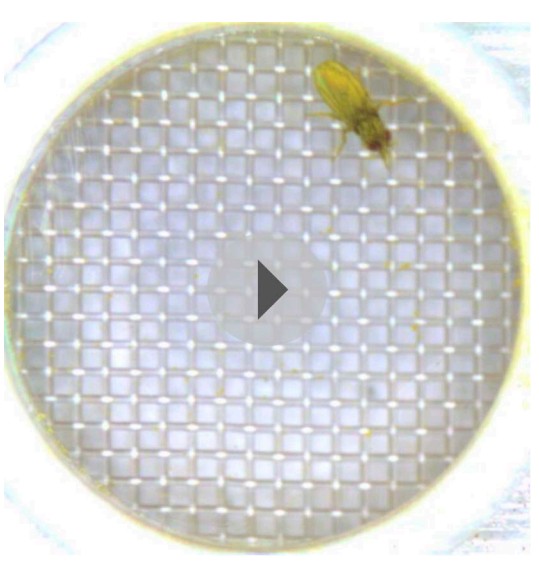

**Video 1**. Cleaning movements of a wild-type fly after being coated in dust. This video is related to *Figure 1*.

cleaning where they left off or transitioned to other body parts (*Figure 1G*). Thus, flies groom by gradually cleaning different body parts in bouts that are characterized by cyclic transitions between body cleaning sweeps and leg rubbing (shown as a series of transition diagrams in *Figure 1—figure supplement 6*). The priority order for cleaning the different body parts is: eyes > antennae > abdomen > wings > thorax.

## Grooming behavior is modular

The modularity of grooming is evident from observations that individual cleaning movements can be evoked by locally applied stimuli (*Vandervorst and Ghysen, 1980*; *Corfas and Dudai, 1989*). This led us to ask whether grooming movements are modular enough to be activated independently in the brain. We expressed the temperature-sensitive cation channel dTrpA1 (*UAS-dTrpA1*) (*Hamada et al., 2008*) in different enhancer-driven GAL4 expression patterns in the nervous system (*Jenett et al., 2012*) and screened for grooming phenotypes (see 'Materials and methods' for details). Using this method to activate neurons targeted by different GAL4 lines, we discovered that grooming movements could indeed be independently activated in the absence of dust. We identified different GAL4 lines that caused exclusive cleaning of each body part, including the head, abdomen, wings, and legs (*Figure 2*, *Figure 2—figure supplement 1*, *Figure 2—figure supplement 2*; see *Video 2*, *Video 3*, *Video 4*, *Video 5*, *Video 6*, *Video 7*, *Video 8*, and *Video 9* for representative videos and more detail regarding the behavioral phenotypes). Some lines we identified drove cleaning of different body sub-parts such as the antennae, in agreement with our observations that flies precisely target cleaning of different regions of the head (*Figure 2*, R26B12). Although these lines were sufficient for driving highly specific cleaning movements, we did not find evidence that blocking the activity of their targeted neurons could disrupt dust-induced grooming (tested using the neural inhibitors *UAS-Shibire$^{ts1}$* or *UAS-TNT*, data not shown). This suggests that the targeted neurons are sufficient but not necessary for driving their respective cleaning movements. All lines tested could activate continuous cleaning of their corresponding body parts for long periods of time (at least 25 min). Thus, grooming can be decomposed into specific motor programs (hereafter referred to as cleaning modules) that can be independently activated by subsets of neurons. Experimental induction of these cleaning modules via GAL4-expressed dTrpA1 allowed us to directly test how they are sequentially executed during the progression of grooming.

## Hierarchical suppression mediates the selection of cleaning movements

We postulated that coating flies with dust leads to the simultaneous stimulation of all the cleaning modules, but because of body mechanical constraints, these competing modules are mutually exclusive and therefore must be performed one at a time. Such temporal conflicts between competing behaviors can be resolved if the performance of one behavior suppresses the others, such that only one is selected at a time (*Davis, 1979*; *Redgrave et al., 1999*; *Kupfermann and Weiss, 2001*; *Kristan, 2008*). Building on this concept, we hypothesized that asymmetries in intermodular suppression could underlie biases in the preferred order of execution. Specifically, we hypothesized that a suppression hierarchy between cleaning modules could mediate fly grooming priorities. In this schema, the ability of the cleaning modules to suppress each other forms a hierarchy in which modules 'higher' in the hierarchy are able to suppress 'lower' modules, but not vice-versa. For this hypothesis to account for the fly grooming progression, the order of suppression within the hierarchy would have to match the order of the normal grooming sequence, as schematized (*Figure 3A*). Thus, in a completely dusted fly, the most hierarchically superior module is able to suppress all the other competing modules, the second highest module can suppress all but the first one, and so on.

Our GAL4 lines allowed us to test this hierarchical suppression hypothesis by artificially driving cleaning of specific body parts while simultaneously dusting the flies to induce competing cleaning movements. For this experiment, we selected lines that could induce a particular cleaning module and lacked additional phenotypes that might confound our grooming-specific interpretations (*Figure 2*, lines labeled in red, *Video 2*, *Video 3*, *Video 4*, *Video 5*, *Video 6*, *Video 7*, *Video 8*, and *Video 9*). *Figure 3B* illustrates the outcome of these experiments, which matched the prediction of the hierarchical suppression hypothesis (compare *Figure 3—figure supplement 1A* to *Figure 3B*). For example, when a fly carrying a GAL4 driver that activates abdominal cleaning (R24B03) was stimulated in the presence of whole-body dust, it first cleaned the head, then abdomen (just as wild-type flies do), but then persisted in cleaning its abdomen instead of proceeding to its wings and thorax (*Figure 3B*,

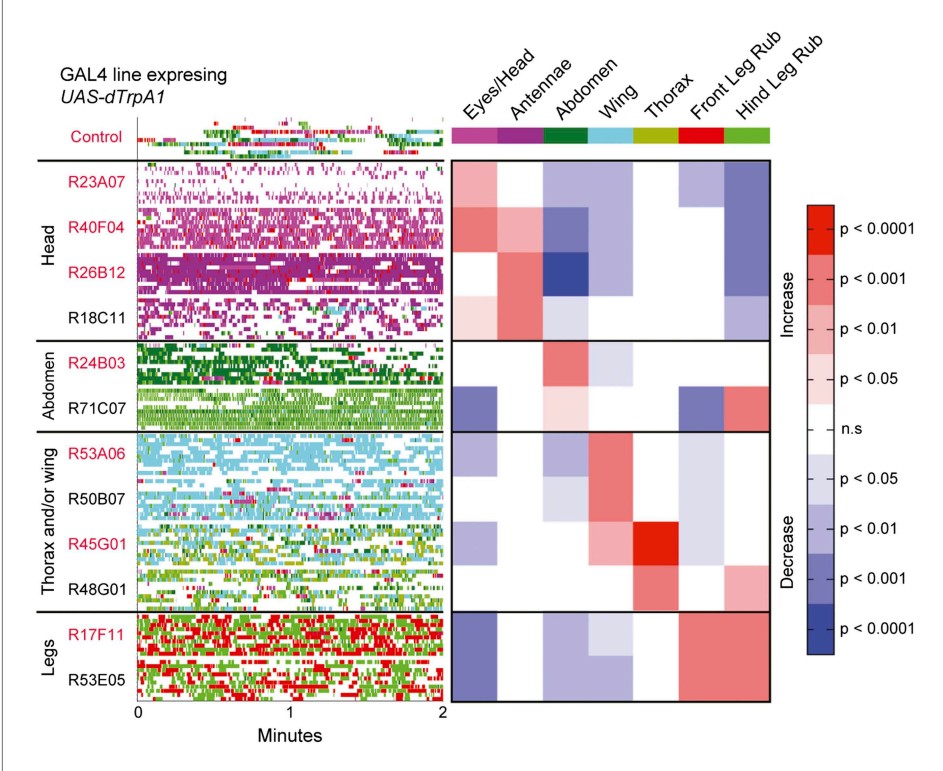

**Figure 2**. Activation of *UAS-dTrpA1* in different GAL4 lines is sufficient to activate discrete cleaning movements in the absence of dust. Cleaning movement activation phenotypes driven by 12 GAL4 lines expressing *UAS-dTrpA1*. Flies (including controls) were warmed to 30°C to activate the targeted neurons while their cleaning movements were recorded and manually scored (n = 10 flies/GAL4 line; 130 flies total). Ethograms of the scored behaviors are displayed by compressing all mutually exclusive events to a single line per fly. Colors below the movement names correspond to those on the ethograms. White space in each ethogram represents the time flies spent walking or standing in place. The GAL4 lines are grouped into four cleaning movement categories: head, abdomen, thorax and/or wings, and legs. The grid displays increases and decreases from control flies in the fraction of time each line spent performing different cleaning movements. Grid heat map represents the p-values for the comparisons of the different GAL4 lines and control flies (Kruskal–Wallis followed by Mann–Whitney U pairwise tests and Bonferroni correction). Note: R71C07 displays significant increases in both abdominal cleaning and leg rubbing. Although this line is shown in the abdominal cleaning category, it could also be included with leg rubbing. Lines labeled in red are used in experiments shown in *Figure 3* and *Figure 5*. See *Video 2*, *Video 3*, *Video 4*, *Video 5*, *Video 6*, *Video 7*, *Video 8*, and *Video 9* for representative videos and more description of the activation phenotypes of these lines.

The following figure supplements are available for figure 2:

**Figure supplement 1**. GAL4 lines expressing *UAS-dTrpA1* have different activated cleaning phenotypes at high temperature.

**Figure supplement 2**. Anatomy of GAL4 lines used to activate distinct cleaning movements.

*Figure 3—figure supplement 2A,D*, see R24B03). This demonstrates that abdominal cleaning is below head cleaning but above wing and thoracic cleaning in the suppression hierarchy. In contrast, flies carrying GAL4 drivers that activate head cleaning (R23A07, R40F04, R26B12) failed to remove dust from their posterior body, consistent with the hypothesis that the behavior would suppress posterior cleaning modules (*Figure 3B*, *Figure 3—figure supplement 1*). GAL4 lines that activated abdominal (R24B03), wing (R53A06), or all posterior cleaning modules (R45G01) showed little impairment in removing dust from their heads (*Figure 3B*, *Figure 3—figure supplement 1*, see R24B03, R53A06, and R45G01). Like wild-type flies, these GAL4 lines cleaned their heads at the onset of dusting,

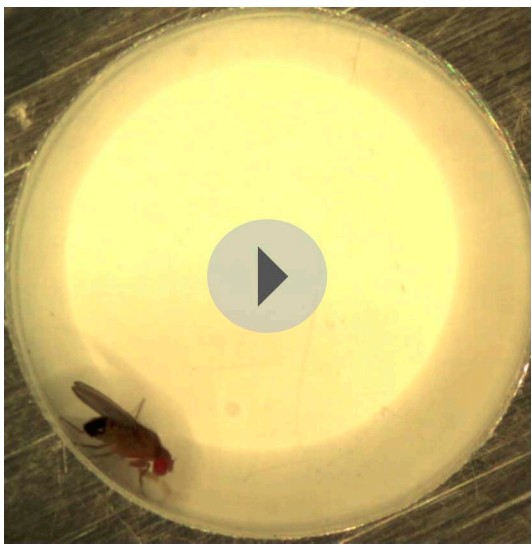

**Video 2**. Activated eye and head cleaning (R23A07-GAL4 / UAS-dTrpA1). This video is related to *Figure 2*. Activated at 30°C. Displayed minor walking defect that was unrelated to the cleaning phenotype.

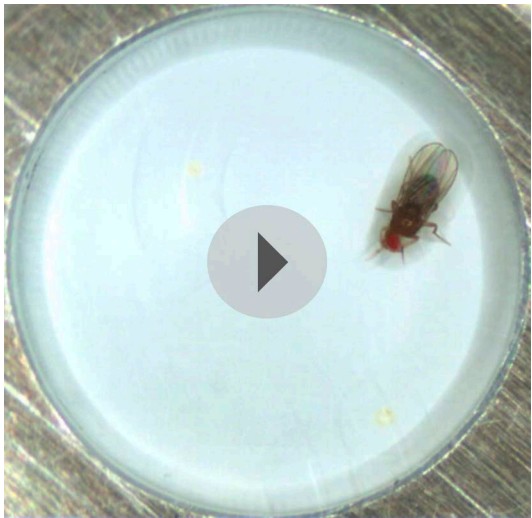

**Video 3**. Activated whole head cleaning (R40F04-GAL4/UAS-dTrpA1). This video is related to *Figure 2*. Activated at 30°C. No other overt phenotypes were observed.

indicating that the normal progression of cleaning was preserved up to the point where it was blocked (*Figure 3—figure supplement 2A,D,E*, see R24B03, R53A06, and R45G01). This demonstrates that the 'endogenously' produced (dust stimulated) head-cleaning module suppressed strong, artificially activated abdominal, wing, and thoracic cleaning. The fact that head cleaning suppresses other movements through both artificial and endogenous activation argues against the possibility that suppression was an artifact of dTrpA1 activation. These results support our hypothesis that the cleaning modules are competing for output via an intermodular suppression that favors those that are hierarchically superior. The full suppression hierarchy of cleaning modules is: eyes > antennae > abdomen > wings > thorax, which mirrors the serial order of grooming observed in dusted flies.

## Sequential activation and termination of cleaning movements is mediated by dust on each body part

Although the priority order of cleaning module execution can be attributed to a suppression hierarchy, the sequential progression of grooming requires the termination of the hierarchically superior cleaning module followed by selection of the next module. One possible explanation for the observed progression is that each module sequentially triggers the next module in the series, analogous to an activation chain (*James, 1890*; *Adams, 1984*). We tested this possibility using the GAL4 lines to elicit different cleaning modules in the absence of dust to see if activation of one cleaning module would stimulate execution of the next module. We found that activation and subsequent acute deactivation of neurons that controlled a given cleaning module did not directly elicit the next module in the grooming sequence (*Figure 3—figure supplement 3A,C*). This indicated that an activation chain among the cleaning modules is not sufficient to direct the progression of grooming. This conclusion is supported by observations that cleaning movements are only directed to the body part that is stimulated and not to other parts (*Vandervorst and Ghysen, 1980*; *Corfas and Dudai, 1989*; *Kays et al., 2014*)

An alternative way by which grooming could progress is through a combination of the acute sensory cues provided by dust and the intrinsic suppression hierarchy among the cleaning modules. To test this hypothesis, we took advantage of the fact that artificial activation of specific cleaning modules blocks grooming of subordinate body parts (*Figure 3B*). Thus, if the cleaning modules are evoked by dust on specific body parts, then flies released from artificial activation should initiate cleaning of the next dirty body part in the hierarchy. We tested this prediction by rapidly cooling the dusted GAL4/*UAS-dTrpA1* flies shown in *Figure 3B* to a temperature at which dTrpA1 is inactive,

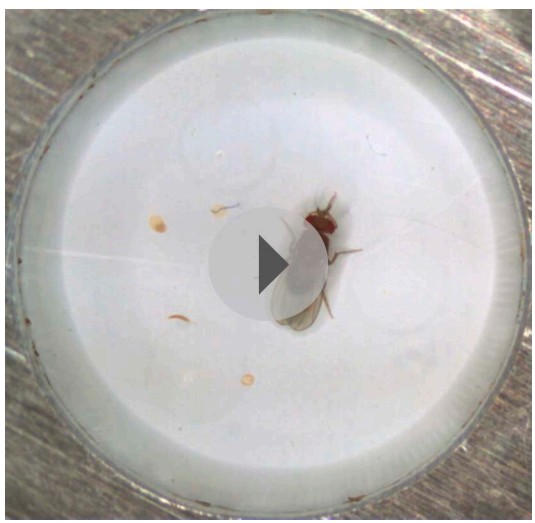

**Video 4**. Activated antennal cleaning (R26B12-GAL4/UAS-dTrpA1). This video is related to *Figure 2*. Activated at 30°C. No other overt phenotypes were observed.

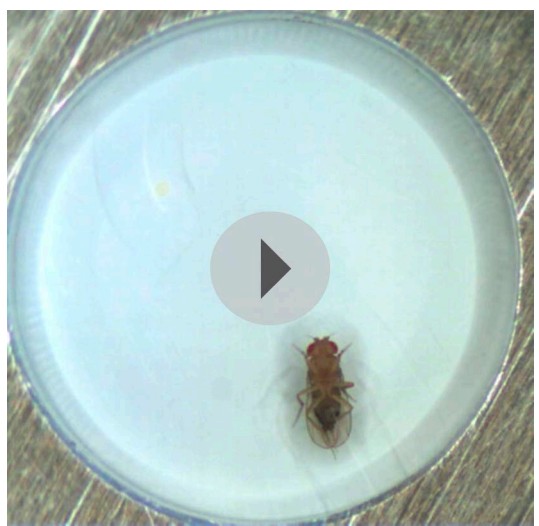

**Video 5**. Activated abdominal cleaning (R24B03-GAL4/UAS-dTrpA1). This video is related to *Figure 2*. Activated at 30°C. No other overt phenotypes were observed.

and then observing the first cleaning movements performed. In effect, flies were returned to a normal behavioral state but with different distributions of dust on their bodies (*Figure 3—figure supplement 3A,C*). Indeed, flies resumed cleaning the body part downstream of where the sequence had been blocked (*Figure 3C*, *Figure 3—figure supplement 3B,D*). For example, flies with clean heads and abdomens, but dirty wings and thoraces, focused on cleaning their wings (*Figure 3C*, line R24B03). Thus, cleaning progresses to the next dirty and hierarchically superior part.

## A parallel model of serial behavior can explain sequential grooming

We found that hierarchical suppression can mediate the selection of one among many simultaneously stimulated cleaning modules that are in competition for output. Between the proposed models describing how nervous systems execute serial actions, hierarchical suppression through competition is a core feature of parallel response models rather than activation chain models (*Houghton and Hartley, 1995*; *Bullock, 2004*). The general schema for this motor-control strategy for grooming is shown in *Figure 4A* and consists of three layers: a sensory layer, a hierarchical layer, and a winner-take-all layer. The sensory layer consists of parallel sensory inputs that each independently activate a specific cleaning module. The hierarchical layer establishes differences in the activation levels between the cleaning modules (described in detail below). The winner-take-all layer then selects the module with the highest activation level by suppressing execution of the other modules. Once a module is selected, the reduced sensory drive that results from cleaning dust from the relevant body part gradually lowers its activation level, thereby allowing cleaning to proceed to whichever module has the next highest activation level (In *Figure 4A*, the removal of dust by a module is represented with blunt arrows from the winner-take-all layer to the dust). Grooming proceeds from the module with the highest activation level to that with the lowest until dust is removed from all of the body parts. This organization leads to the emergence of sequential cleaning of body parts.

The general strategy of hierarchical suppression can be implemented using two related schemata. We simulated both possible implementations with a computational model to test if they were sufficient to explain the progression of grooming that we observed. The winner-take-all layer is the same for both implementations. Winner-take-all neural networks can be modeled using all-to-all inhibitory connections between the different units; however, conceptually, they simply select the most active unit and inhibit the rest (*Houghton and Hartley, 1995*). In the model presented here, we selected the cleaning module with the highest activation level by using a maximum function ('Materials and Methods'). The only difference between the two implementations of the model is how the activation

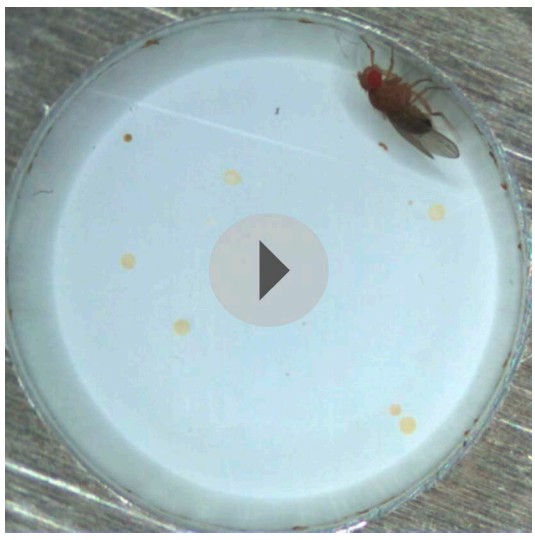

**Video 6**. Activated wing cleaning (R53A06-GAL4/
UAS-dTrpA1). This video is related to *Figure 2*.
Activated at 30°C. No other overt phenotypes were
observed.

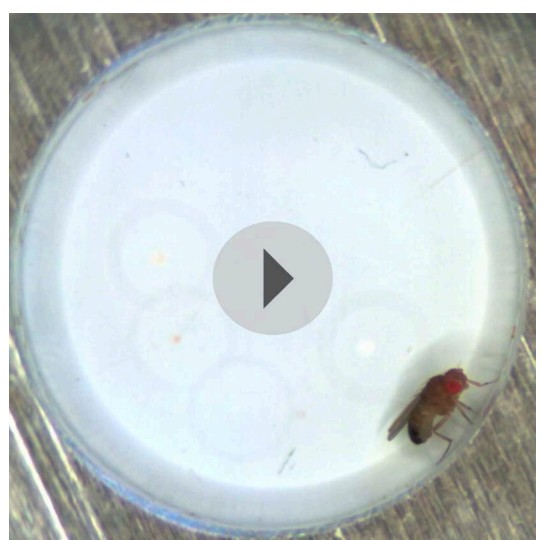

**Video 7**. Activated posterior body cleaning (R45G01-
GAL4/UAS-dTrpA1). This video is related to *Figure 2*.
Activated at 30°C. No other overt phenotypes were
observed.

level of each cleaning module is set in the hierarchical layer. If no differences are set in the relative activation levels between modules, the result is unordered grooming, with equal probability of the modules occurring (*Figure 4B*). Alternatively, the sensitivity to dust (sensory gain) across the modules can be varied, such that their activation levels are proportional to the amount of dust on each body part, weighted by their sensitivity to the dust stimulus (gain of response to the dust stimulus). With this schema, modules with the highest sensory gain had a higher probability of being selected than those with lower gains, reproducing the observed sequence of grooming behavior (*Figure 4C*). We see a similar result when the differences in the activation levels of the competing modules arise from unidirectional (asymmetric) lateral inhibitory connections (*Figure 4D*). Thus, both simulations produce the sequential selection of cleaning modules observed in grooming behavior. This demonstrates that hierarchical suppression can be established through either variable sensory gain or direct unidirectional inhibition, in conjunction with winner-take-all selection of the module with the highest activation level.

The model of hierarchical suppression also replicates the returns to previous cleaning modules that we observed with dusted flies (*Figure 1E*). Based on the computational model, this occurs when the cleaning modules have relatively similar activation levels. For example, when no hierarchical differences are set among the modules (i.e., *Figure 4B*), they are all activated to nearly the same level in response to dust, and their selection is dependent only on slight differences in dust levels on their corresponding body parts. Once selected, a cleaning module reduces dust on its body part, causing its activation level to drop below that of the next cleaning module. That next module is then selected and cleans the associated body part until its own activation level falls below that of another module, and so on. Thus, incremental amounts of dust are removed from body parts while frequent switching occurs among the different modules (*Figure 4B*). In the case where a hierarchy is set among the cleaning modules (i.e., *Figure 4C–D*), the selection of those that are most hierarchically superior is initially very probable because their activation levels are relatively high in response to the starting amount of dust. As dust is removed, however, these levels among the modules become more similar. This equalization of activation levels through dust removal results in return cleaning by the modules, and the simulations shown in *Figure 4C,D* come to more closely resemble that of *Figure 4B* as cleaning progresses. Such return cleaning is also seen in wild-type grooming flies, and can thus be explained by our parallel model (*Figure 1E*).

In addition to simulating the grooming progression, the model recapitulates two other important features of our experimental data. First, it can mimic the constitutive cleaning phenotypes of GAL4

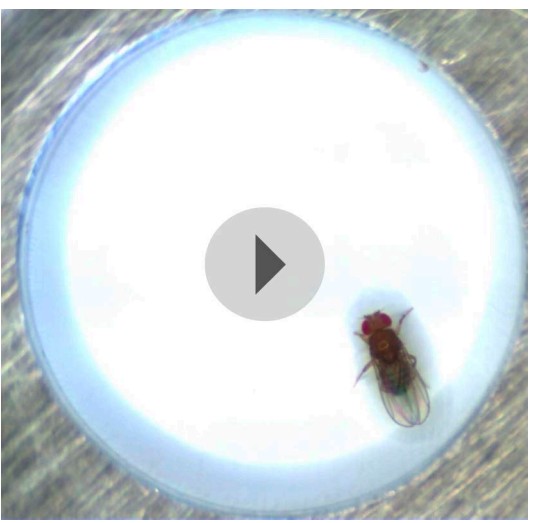

**Video 8**. Activated leg rubbing (R17F11-GAL4/ UAS-dTrpA1). This video is related to *Figure 2*. Activated at 30°C. No other overt phenotypes were observed.

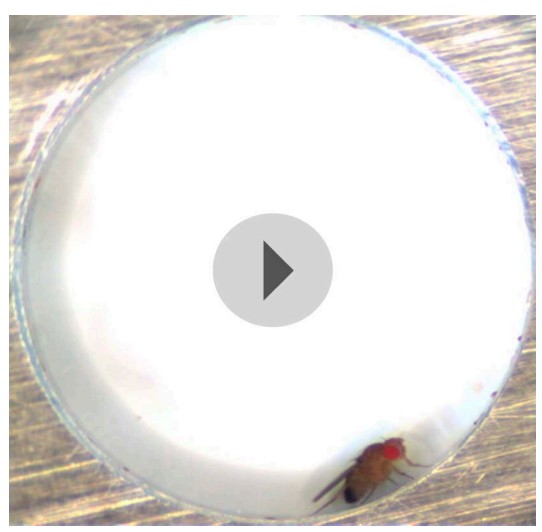

**Video 9**. Control for activation experiment (Control/ UAS-dTrpA1). This video is related to *Figure 2*. Activated at 30°C. No phenotypes were observed.

lines expressing dTrpA1 in the absence of dust. Constitutive activation was implemented by setting a particular body part to be fully dusted regardless of the amount of cleaning done (as if the sensory neurons were constantly active). This resulted in the continual cleaning of a particular body part without triggering any other cleaning modules (*Figure 4E*). Second, the model recapitulates the hierarchical suppression that we observed in our competition experiments (shown in *Figure 3A*). Adding dust to all body parts in the presence of the constitutive activation of a module and unidirectional inhibitory connections simulated hierarchical suppression. This leads to sequential cleaning by the hierarchically superior modules, whereas cleaning by the inferior modules was absent, despite the presence of dust (*Figure 4F*). Thus, we find close agreement between the experimental results and the model of grooming behavior, both in the progression of cleaning modules down the hierarchical levels and in the ability of modules higher in the hierarchy to suppress the modules below.

## Hierarchical suppression mediates cyclic transitions between body cleaning and leg rubbing

Our model explains how the sequential selection of cleaning modules occurs, but could it also explain the frequent alternations between cleaning the body parts and leg rubbing that occur throughout the grooming progression? These alternations are an important and prominent feature in normal grooming behavior for efficient dust removal (*Figure 1E,G*, *Figure 5A*). In order to simulate the alternations, we considered leg rubbing to be a member of the same suppression hierarchy with the other cleaning modules and ran the model simulation based on two premises (*Figure 5B*). First, leg rubbing occurs at all stages of grooming. Therefore, it would necessarily be among the most hierarchically superior of the cleaning modules to be able to suppress cleaning by the others. Second, dust accumulates on the legs while they clean the body parts. This accumulation of dust

increases the activation level to the point where leg rubbing is active and the body cleaning modules are subsequently suppressed. The removal of dust from the legs through leg rubbing then reduces suppression on the body cleaning modules and allows cleaning of the corresponding body parts to resume. This cyclic accumulation and removal of dust from the legs during body cleaning could account for the high transition probabilities between the leg and body modules. When we simulated leg rubbing in the model, the sequential progression of body cleaning modules down the hierarchy was preserved (*Figure 5C*). The simulation also produced the cyclic alternations between cleaning of a body part and leg rubbing that we observed in our empirically obtained ethograms of wild-type flies (*Figure 5C,D*). We conclude that hierarchical relationships among the cleaning modules mediate not only their sequential execution, but also the cyclic transition dynamics between body cleaning and leg rubbing.

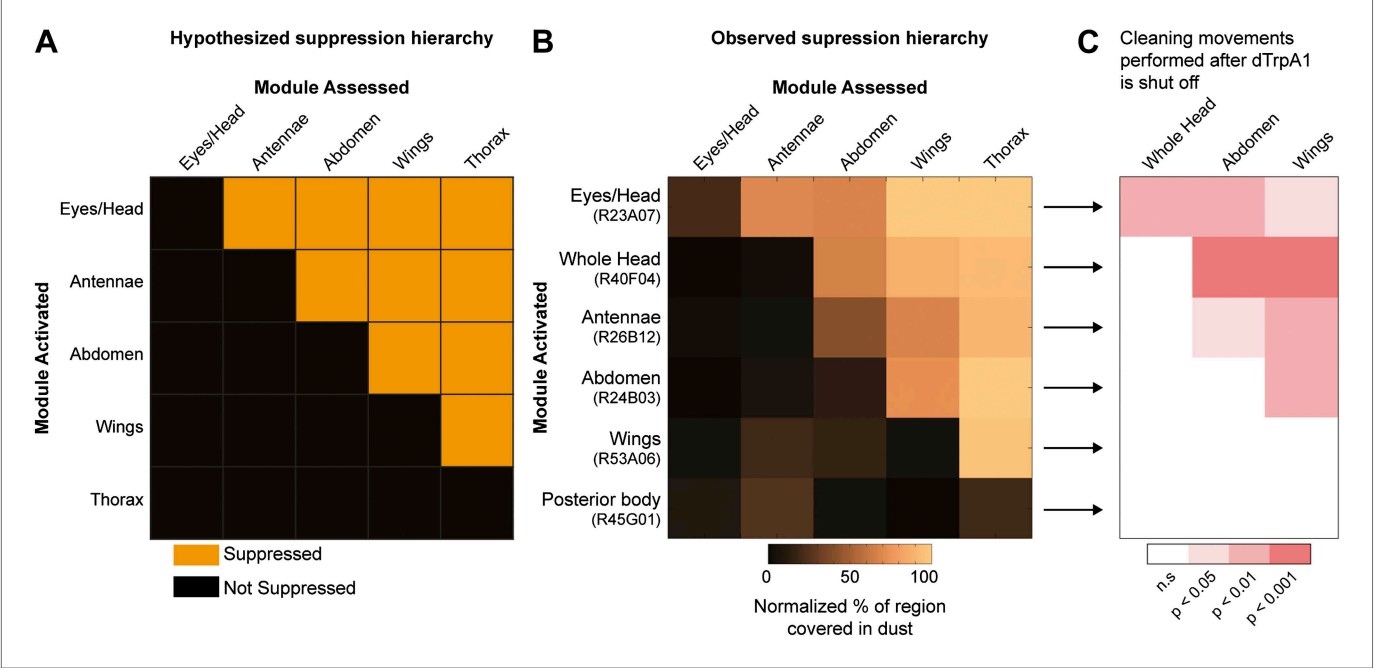

**Figure 3**. Hierarchical suppression and dust stimulus drive cleaning movement selection. Cleaning of specific body parts was artificially activated while flies were dusted to stimulate competition between their cleaning movements. Flies were pre-warmed at 30°C such that the dTrpA1-induced cleaning module was active at the time of dusting. After grooming for 25 min, flies were anesthetized and their dust patterns were measured. (**A**) Grid showing the expected suppression pattern if the hierarchical suppression hypothesis is true. Modules are arranged on the grid in the order that they occur in the normal grooming sequence. (**B**) The observed suppression hierarchy. For each line, the normalized fraction of dust remaining on different regions of the flies is mapped onto the corresponding grid locations (n ≥ 26 per body part, 'Materials and methods'). The module activated by each GAL4 line is listed above the line name. Data used to generate the grid is shown in *Figure 3—figure supplement 1*. (**C**) Cleaning movements performed when a GAL4/dTrpA1-activated module is shut off. Arrows from **B** to each row in **C** show the GAL4 line and corresponding dust distribution that was tested. The grid displays increases from control flies in the frequencies of different cleaning movements performed in the first 3 minutes after shutting off dTrpA1 (n = 10 flies per line). Grid heat map represents the p-values for the comparisons of the different GAL4 lines and control flies (Kruskal–Wallis followed by Mann–Whitney U pairwise tests and Bonferroni correction). Movements were manually scored. All head cleaning movements are binned and displayed as whole head, because eye and antennal cleaning are not easily distinguishable in the dusted state. Control and experimental flies performed few thoracic cleaning bouts and are therefore not shown.

The following figure supplements are available for figure 3:

**Figure supplement 1**. Dust patterns resulting from coating flies in dust and artificially activating specific cleaning movements.

**Figure supplement 2**. Behaviors of flies that were coated in dust while specific cleaning movements were artificially activated.

**Figure supplement 3**. Triggering of cleaning movements is dust dependent.

Our model predicts that leg rubbing is among the most hierarchically superior of the cleaning modules. Therefore, if leg rubbing is constitutively activated, it should suppress cleaning of the other body parts in a competition experiment where flies are coated in dust. We tested this prediction by using a GAL4 line that drove strong leg rubbing of both the front and hind legs (*Figure 2*, R17F11). When these flies were coated in dust, they were unable to completely clean their bodies, indicating that the body cleaning modules were suppressed (*Figure 5E*, *Figure 3—figure supplement 1B*). This supports the prediction that leg rubbing is part of a grooming suppression hierarchy. However, cleaning of some body parts were suppressed more than others, suggesting that leg rubbing is not the most superiorly positioned module in the suppressive hierarchy. For example, activated leg rubbing did not suppress eye cleaning. This suggests that eye cleaning is at the top of the grooming hierarchy. Additionally, leg rubbing only weakly suppressed cleaning of the abdomen. Given that abdominal cleaning is the most hierarchically superior of the cleaning modules performed by the hind legs, this may suggest that hind leg rubbing and abdominal cleaning are more equal in their positions in the hierarchy. These results

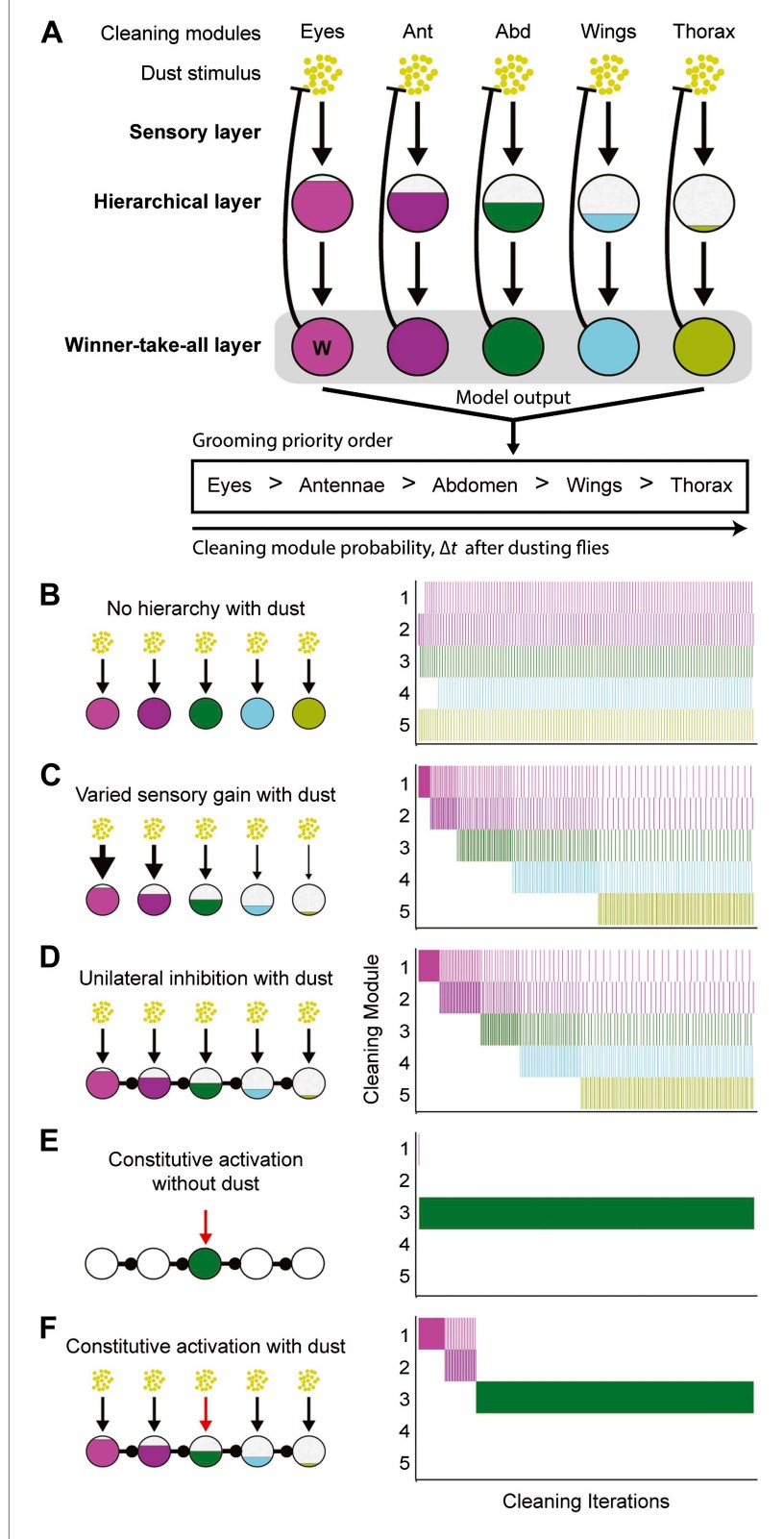

**Figure 4**. Model of hierarchical suppression results in the sequential progression of grooming. (**A**) The dust induced grooming sequence requires three layers: (1) the sensory layer detects dust and independently activates each cleaning module. This is shown as parallel excitatory arrows from each yellow dust cartoon to activate specific

*Figure 4. Continued on next page*

*Figure 4. Continued*

cleaning modules. (2) The hierarchical layer determines each module's level of activation when its respective body part is coated in dust. Circle fill levels show theoretical differences in the relative activation levels of the modules. (3) The winner-take-all layer selects the cleaning module that is most active and suppresses all competing responses ('W' in this layer shows that the eye cleaning module is selected first). Theoretical all-to-all inhibitory connections in this layer are depicted as a gray box for simplicity. Blunt arrows from the winner-take-all layer to the yellow dust depict that the winning module reduces its own sensory input by cleaning off the dust and consequently becoming less active. The cleaning continues until the activation level of the module is no longer maximal, at which point the transition of cleaning to the new maximally active module occurs. Multiple iterations of this process result in a sequential progression. (**B–D**) Computational model simulates possible implementations of the hierarchical layer in establishing the most active modules. Modifications to the hierarchical layer and sensory inputs are depicted in each diagram. In this simulation, the competition is between five different cleaning modules. The ethograms show typical results of the simulation, where each row corresponds to the output of a module. (**B**) Equal sensitivities to dust and no inhibitory connections. (**C**) Modules with varying sensory gain in response to dust: modules with higher sensory gain (depicted with thicker arrows) have higher activation levels in response to the same dust amount. Fill levels represent the relative activity levels of the modules at the first iteration of the simulation. (**D**) Unidirectional lateral inhibitory connections between the modules. For simplicity of illustration, only the nearest-neighbor inhibitory connections are shown; in the computational implementation, each module inhibits all the subordinate modules in the hierarchy (e.g., 2 inhibits 3, 4, and 5). (**E**) Constitutive activation of a single cleaning module. Simulated by setting the amount of dust on a particular body part to completely dirty after each round of cleaning (depicted with the red arrow). (**F**) Constitutive activation of a single cleaning module in the presence of dust on all body parts.

---

indicate that our model can explain the main observed features of the grooming progression, including the sequential cleaning of body parts and the frequent alternations between leg rubbing and body part cleaning.

## Discussion

### Hierarchical suppression and its relationship to behavioral choice

Hierarchical associations among different available behavioral options allow for those behaviors with the highest value for a given circumstance to be selected while all others are suppressed. These concepts are often referred to as behavioral choice, action selection, motor program selection, or decision making (*Davis, 1979*; *Redgrave et al., 1999*; *Kupfermann and Weiss, 2001*; *Kristan, 2008*). For example, mollusks will suppress mating in order to feed (*Davis, 1979*). This suggests that food is rarer than potential mates, and thus feeding has a higher value than mating. We find that a suppression hierarchy mediates the selection of competing cleaning movements that cannot be executed at the same time. The existence of this hierarchy suggests that inherent values are placed on cleaning the different body parts and selecting one over others represents a 'best' choice. Eye cleaning is executed first in response to dust and is the most hierarchically superior cleaning module. This suggests a high value for flies to keep their eyes clear of debris. Because clean legs are necessary for cleaning each body part, their position near the top of the hierarchy is necessary for efficient debris removal. The antennae may have high value to flies because of the number of different sensory modalities that they house (e.g., audition, olfaction, gravity sensing, wind detection) (*Laissue and Vosshall, 2008*; *Kamikouchi et al., 2009*; *Yorozu et al., 2009*; *Böröczky et al., 2013*; *Wilson, 2013*). Keeping the abdomen clear of debris is likely critical for respiration as it contains numerous spiracles (*Phillis et al., 1993*; *Heymann and Lehmann, 2006*). Because of these different values on behavioral motor programs, animals have evolved mechanisms for selecting the most favorable option and suppressing others.

A significant finding of this work is the experimental demonstration that hierarchical suppression among many behaviors leads to the emergence of a sequence. Although much attention has focused on the role of hierarchical suppression in selecting between behavioral choices, there is little experimental evidence to support its role as a mechanism for eliciting action sequences. The grooming sequence may be considered as a series of 'forced choices' among different competing cleaning modules that are selected in priority order through suppression. As each body part is cleaned, suppression of the other cleaning modules is lifted, and a new round of competition between the remaining cleaning options occurs. Our work lends strong support to an emerging view that hierarchical

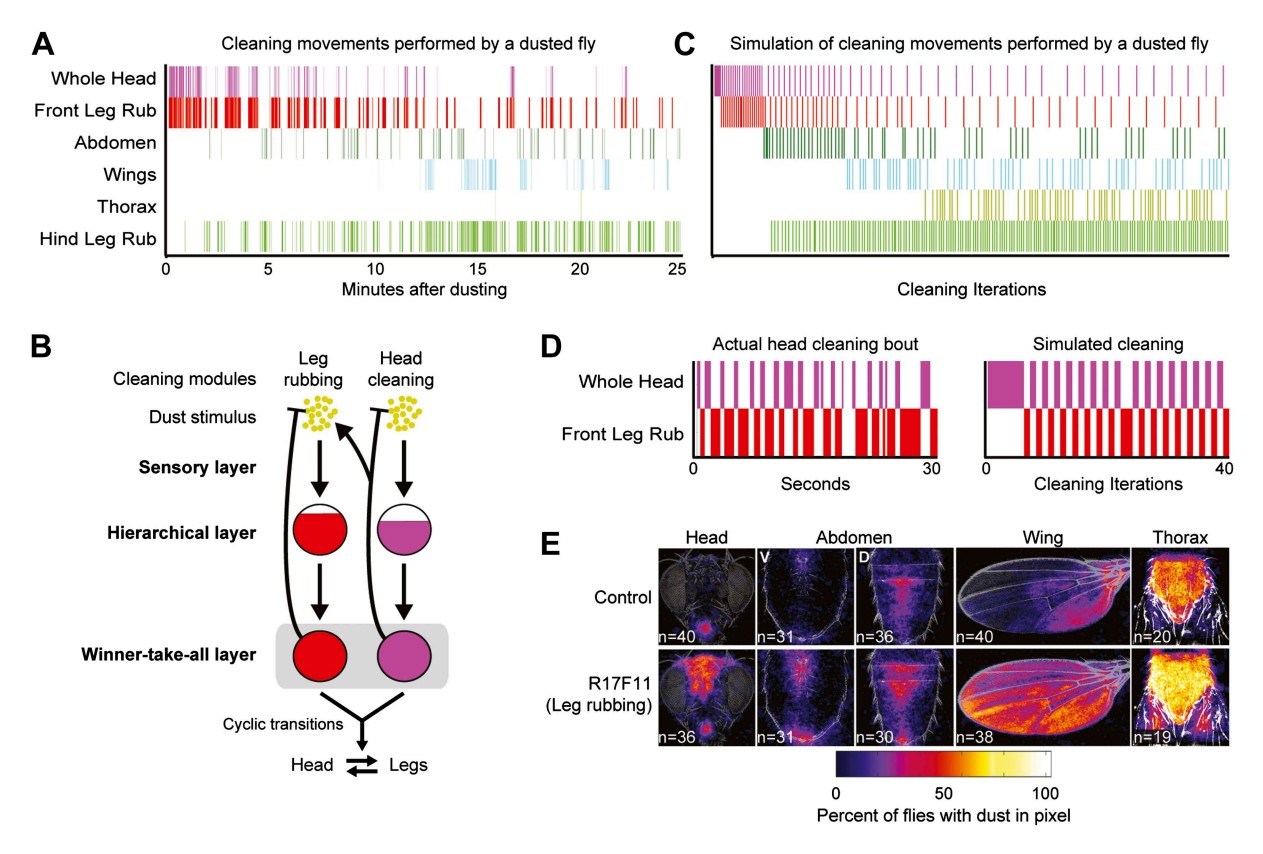

**Figure 5**. Hierarchical suppression mediates the cyclic transitions between cleaning modules. Leg rubbing was simulated in the grooming model based on two features. (1) The legs accumulate dust as they remove it from the body parts. Leg rubbing is subsequently executed to remove that dust. (2) The sensory gain on the legs was set high relative to the other cleaning modules such that they are the most active and selected in the winner-take-all layer when they were sufficiently dirty. (**A**) Ethogram example of a wild-type fly grooming for comparison to the simulation. (**B**) Model of the leg rubbing and body part cleaning cycle (head cleaning example shown). Leg rubbing is hierarchically associated with the body-cleaning modules (similar to the associations among the cleaning modules described in *Figure 4*). The only difference between leg rubbing and the other modules is the accumulation of dust on the legs during body part cleaning. This is depicted by a forked connection that removes dust from the head (blunt arrow) and transfers it to the legs (arrow to the leg rubbing module). (**C**) Simulation of grooming with leg rubbing and four body-cleaning modules (result is typical). (**D**) Typical examples of cyclic switching between body cleaning and leg rubbing. The ethogram on the left displays a wild-type fly cleaning its head and the one on the right shows a simulated head-cleaning bout (each example is from early time points in **A** or **C** respectively). (**E**) Average spatial distribution of dust on each body part that remains when flies were coated in dust while leg rubbing was activated (25 min after dusting). V = ventral, D = dorsal.

suppression is a ubiquitous action selection mechanism, whether it is used in deciding among available behavioral choices, or selecting movements performed in complex serial tasks. Thus, the evolutionarily ancient ability to select among competing behaviors through hierarchical suppression may have provided the necessary infrastructure to perform motor actions in sequence (*Houghton and Hartley, 1995*).

## Relationships between fruit fly grooming and parallel models of serial behavior

Activation chain and parallel models describe two different mechanisms for generating sequential behavior. Activation chains (sometimes referred to as associative chaining, response chaining, or stimulus-response reflex chaining) are the simplest form of sequential execution. These work through feedforward activation of the next movement by the previous one (*James, 1890*). Parallel models (also known as competitive queuing) select movements through winner-take-all inhibition among the different motor units that are competing for output (*Houghton and Hartley, 1995*; *Bullock, 2004*). Thus, the primary difference between the models is movement selection through either feedforward excitatory or competitive inhibitory mechanisms.

Two lines of evidence argue against a primary role for an activation chain in driving the dust-induced grooming sequence. First, each cleaning movement can be activated by local stimulation of its respective body part without triggering other movements (*Vandervorst and Ghysen, 1980*; *Corfas and Dudai, 1989*; *Kays et al., 2014*). Either local stimulation (*Figure 3C*) or neuronal activation (*Figure 3—figure supplement 3C*) of a particular body part-cleaning module activates only one module and does not trigger the others. Feedforward activation models predict a chain of different cleaning modules once the sequence is triggered rather than just one (*James, 1890*; *Rhodes et al., 2004*). Second, the observed structure of grooming behavior in fully dusted flies is inconsistent with an activation chain model. As flies progressively groom their bodies, the transitions among different cleaning modules are gradual (*Figure 1D,E*, *Figure 1—figure supplement 4B*, *Figure 4B–D*). These gradual transitions are characterized by return cleaning among the modules, meaning that previous cleaning modules are revisited after later modules were already selected. Cleaning modules elicited through feedforward activation are not likely to exhibit such return cleaning, but should produce more deterministic sequences, such as those observed for zebra finch bird song (*Long et al., 2010*). Although it is possible that activation chains mediate some aspects of grooming behavior, our evidence indicates that a parallel model alone is fully sufficient to reproduce major features of the dust-induced grooming sequence (discussed below).

We find that the grooming sequence has features that are similar to parallel models of serial behavior. Both involve the parallel activation of premotor units in the sequence. The activation of the cleaning modules occurs through dust sensing, where each module is independently stimulated by a dust stimulus on its corresponding body part. Coating the body of the flies in dust leads to the parallel stimulation of the modules, and thus competition among them. Parallel models resolve such simultaneous activation schemes by creating a gradient of activity levels among the units that are executed in the sequence and selecting the one with the highest activity in a winner-take-all network. Our computational model of grooming works through a similar mechanism (*Figure 4A*). We find that either differences in the sensory gain of the dust response or unidirectional (asymmetric) lateral inhibitory connections among the modules can generate an activity gradient. The most active cleaning module is selected in the winner-take-all layer while the others are suppressed. Parallel models propose that the sequence progresses when the selected unit is deleted, and a new round of competition and suppression occurs between the remaining units. In grooming, the active module 'deletes' itself from the sequence by reducing the dust stimulated sensory drive through cleaning (*Figure 4A*). Thus, all of the major features of parallel models are present in our model of sequential grooming behavior (parallel activation, competition, activity gradient, winner-take-all selection, and deletion).

The finding that grooming behavior displays both sequential and cyclic transitions between modules offers an extension to our understanding of the flexibility of parallel models in producing different types of serial behavior. We found that fruit flies make frequent transitions between body cleaning and leg rubbing and that these oscillations can be explained by the same hierarchical rules as those that govern the sequential progression of the behavior (*Figure 5*). The only difference between leg rubbing and body cleaning is that the act of cleaning the body part leads to the transfer of dust to the legs, which then activates leg rubbing (*Figure 5D*). The cyclic transfer and removal of the dust input leads to oscillatory behavior. Thus, simple modifications to the infrastructure underlying parallel models could produce different types of transition dynamics between modules (i.e., sequential progression and oscillatory dynamics). This demonstrates that these models can generate behaviors of increasing adaptive complexity.

Does fruit fly grooming behavior provide insight into the organization of more complex motor sequences? Our model of the grooming progression has features like those proposed to explain learned sequences that also require parallel activation of units that compete for motor output. An example of this was derived from studies of human typing that revealed that adjacent letters in a word were often exchanged in error. This suggested that a plan for all the letters necessary to generate a word was readied prior to typing and that execution of each letter competed for output based on a hierarchical prioritization scheme and winner-take-all competition (*Lashley, 1951*). Such studies led to the proposal of parallel models as a way nervous systems implement sequential behavior (*Houghton and Hartley, 1995*; *Bullock, 2004*). To our knowledge, work presented here provides the first evidence of a parallel model in an innate behavior and suggests that this organization may be more widespread than previously thought. Putting these concepts together with those discussed about behavioral choice leads to the consideration of the possibility that the neural infrastructure driving

simple behavioral choice has served as a framework for the evolution of complex sequential behaviors such as typing (*Houghton and Hartley, 1995*). The unifying feature of these vastly different examples of action selection is hierarchical suppression among competing actions. This raises the possibility that the underlying neural circuits have common organizational features.

## Behavioral models inform circuit studies

In this study, we took advantage of the modularity of grooming to interrogate its behavioral organization. We gained the experimental control necessary for probing this organization by identifying GAL4 lines that target neurons whose activation drove specific cleaning modules. We show cause and effect experiments indicating that suppressive interactions among the modules govern their selection. Many behaviors are thought to consist of subdivisions of simpler motor programs. This approach of acutely manipulating subcomponents of behaviors and assessing how they affect the other components may prove useful for exploring the organization of other complex motor behaviors.

Our results do not address the neural circuits that mediate the grooming progression but rather suggest possible types of circuit organization that could underlie its sequential nature. Many complex behaviors are likely to involve hundreds or even thousands of neurons and information about the behavioral organization helps focus circuit-mapping efforts on specific features. For example, one prediction of our model of hierarchical suppression points to unidirectional lateral inhibitory connections among the cleaning modules (*Figure 4D*). Previous work has demonstrated that such inhibition can occur through direct inhibitory connections from hierarchically superior behavioral circuits to subordinate competing circuits (*Kovac and Davis, 1977*). Therefore, it may be reasonable to search for direct inhibitory connections among the cleaning motor programs. Alternatively, such inhibition among behavioral motor programs may be due to shared neural circuits. In this case, the activity of a behavioral motor program suppresses another, solely because the shared circuit components cannot drive two behaviors at once (*Briggman and Kristan, 2008*). Future experiments will address whether these or other neural circuit mechanisms govern hierarchical inhibition in serial grooming. Thus, the work presented here provides not only a framework for future experiments to understand how neurons and neural circuits mediate serial grooming, but also a basis for extrapolation to analogous organizational logic in other network-driven sequences.

## Neurons that activate cleaning modules

The GAL4 lines identified in this study target neurons that activate specific cleaning modules without inducing other behavioral phenotypes. What types of neurons might these GAL4 lines target? Two types that are already implicated in initiating movements are sensory and command neurons. Sensory neurons can activate specific motor programs as the initiators of stimulus-response circuits (*Sherrington, 1906*; *Huston and Jayaraman, 2011*). In fruit fly grooming, mechanosensory neuron-coupled bristles can trigger cleaning responses with mechanical stimulation (*Vandervorst and Ghysen, 1980*; *Corfas and Dudai, 1989*). This suggests that activation of these sensory neurons with dTrpA1 could trigger the specific cleaning modules. Command neurons have been defined as interneurons that are necessary and sufficient for activating specific motor programs (*Kupfermann and Weiss, 1978*). However, the neurons targeted by the GAL4 lines used in this study only show sufficiency for activating cleaning movements as expression of *UAS-Shibire^{ts1}* or *UAS-TNT* were unable to block cleaning (unpublished data). This may be explained if the GAL4-targeted neurons are those referred to as 'command-like' (also known as command systems, decision neurons, or higher order interneurons) (*Kupfermann and Weiss, 2001*; *Kristan, 2008*). These neural types are sufficient to activate particular motor programs but do not necessarily fit the strict criteria of also being necessary. Thus, we propose that sensory neurons, command-like neurons, or both activate the cleaning modules.

The expression patterns of the GAL4 lines in this study vary in the numbers and types of cells that they target, raising the question as to which neurons in each pattern activate the cleaning modules (*Figure 2—figure supplement 2*). It is likely that the lines target a mixed population of neurons, where only some are grooming specific. There are examples of GAL4 lines that activate specific behaviors, where only a subset of the neurons within each expression pattern confers the phenotype (*von Philipsborn et al., 2011*; *Luan et al., 2012*). Alternatively, the enhancer fragments used in generating the GAL4 lines might target expression to many different types of neurons involved in the same behavior (*Pfeiffer et al., 2008*). For example, the *fruitless* gene is expressed in 1500 neurons of different types that comprise elements of a courtship circuit (*Yu et al., 2010*). Activation of this population of

*fruitless* neurons with dTrpA1 is sufficient to trigger courtship behavior (*Pan et al., 2011*). Further refinement of the expression patterns driven by the GAL4 line enhancers will allow us to distinguish between these possibilities. Thus, we anticipate that identifying the neural circuitry for each cleaning module and determining how these circuitries interact will reveal how the serial organization of grooming behavior is encoded in the nervous system through a hierarchical suppression mechanism.

## Materials and methods

### Fly stocks and rearing conditions

Canton S flies were obtained from Martin Heisenberg's lab in Wurzburg, Germany. GAL4 stocks were generated in Gerry Rubin's lab at Janelia Research Campus by methods described previously (*Pfeiffer et al., 2008*) and are available at the Bloomington stock center (*Jenett et al., 2012*). BDPGAL4U was used as a control for the GAL4 lines in our experiments and contains the vector backbone used to generate the GAL4 collection (including GAL4) but lacks an enhancer to drive its expression. *UAS-dTrpA1* (on the second chromosome) was obtained from Paul Garrity at Brandeis University (*Hamada et al., 2008*). 20x*UAS-mCD8::GFP* (JFRC7) was constructed as described in *Pfeiffer et al. (2010)*. GAL4 lines were crossed to their respective UAS drivers, and both the parents and their progeny were maintained using the following conditions. Flies were reared at 21–22°C and 50% relative humidity using a 16/8 hr light/dark cycle on standard cornmeal and molasses food. All experiments were done with 5- to 8-day-old males.

### Grooming apparatus

*Multiwell grooming chambers*: blocks of grooming chambers were designed to facilitate the parallel and uniform dusting of flies for dust pattern measurements and recording videos of cleaning behavior. Blocks were fabricated with 24 or 96 grooming chambers (15.6 mm or 7 mm diameters respectively) and have connector ends and dust removal ends. The connector end has tracks for thin aluminum sliders that gate each chamber to facilitate fly loading (*Figure 1—figure supplement 1*). The dust removal end consists of a Nitex mesh-covered opening that allows free dust to exit the chamber while preventing flies from escaping (Genesee Scientific Corporation, San Diego, California, 630 μm mesh). Chambers were printed on a 3D printer. *24-well aluminum chambers (for cooling experiments)*: aluminum chambers were identical to those described above, with the exception that the dust removal end was closed such that flies would stand directly on the aluminum (shown in *Figure 1—figure supplement 1E,F*). These chambers were fabricated for rapid cooling of flies to quickly shut off dTrpA1-activated behaviors. Design plans for the grooming chambers are available in *Supplementary file 1*.

### Dusting flies for body surface imaging or video recordings

All experiments were carried out in a warm room that was set to 30–32°C. Flies were cold anesthetized (1–2°C), transferred to grooming chambers (single fly per chamber), and allowed to recover for at least 15 min before assaying. Flies were tapped from the chambers into multiwell plates containing measured amounts of Reactive Yellow 86 dust (Organic Dyestuffs Corporation, Concord, North Carolina) and shaken to uniformly coat each fly (*Phillis et al., 1993*). Excess dust was removed by tapping the flies against the mesh on the dust removal end of the grooming block. The flies were then allowed to clean in the chambers with the mesh side down to allow dust to exit. Video recordings of flies were made with a clear acrylic plug covering the adaptor end of the chamber (*Figure 1—figure supplement 1D*). Grooming was terminated at different time points by anesthetizing the flies with $CO_2$. 0-min time samples were obtained by dusting the flies and immediately anesthetizing them. Data were acquired from at least four separate crosses. 25 min was chosen as the assay end because most body parts were clean at that point.

### Quantification of body surface dust patterns

Dusted flies were prepared as described above and dissected for imaging of their dust patterns. The dissection procedure made it impossible to image the entire body surface of a single fly. Each part was dissected as described below. *Heads:* flies were decapitated using a razor blade, and the heads were affixed face up to a cover slip with double-sided tape (3M Corporation, St. Paul, Minnesota). *Wings:* the left (lateral) wings were removed by grasping the base with a pair of forceps and pulling it off with an adjacent pair of forceps. They were affixed dorsal side up to a cover slip with double-sided tape. *Ventral abdomen:* dissected and imaged shortly after performing grooming assays because the abdomens desiccated quickly. Flies were held on their sides with a scalpel inserted between the head and thorax while all legs and wings were removed. The scalpel was then used to decapitate the fly. Next,

the thorax and dorsal abdomen were glued to a cover slip such that the ventral abdomen was facing up using Elmer's No-Wrinkle Rubber Cement (Elmer's Products, Inc., Columbus, Ohio). *Dorsal abdomen:* flies were held on their sides and the ventral side of the thorax and legs were severed from the dorsal side along a plane that was parallel to the abdomen. The wings were removed and the abdomen was glued dorsal side up. *Thoraces:* no dissection was necessary for imaging thoraces. They were imaged in specially designed chamber that would hold all flies uniformly dorsal side up. Standard preparations and example dirty samples for each body part are displayed in *Figure 1B*.

Body parts were imaged using a Zeiss SteREO Discovery.V12 equipped with an Achromat S 1.0× objective (Carl Zeiss Corporation, Oberkochen, Germany). Heads, abdomens, wings, and thoraces were imaged at 80×, 60×, 65×, and 60× respectively. Z-series were taken and the images were converted into a single, in focus, image with Zeiss AxioVision software using the Extended Focus module (*Figure 1B*, *Figure 1—figure supplement 2A*). Experimental images were manually warped to standard images (see standards and experimental examples *Figure 1B*) using the Photoshop transform tool to ensure that the coordinates on each image corresponded to the same location (Adobe Systems Incorporated, San Jose, California). Yellow dust patterns were isolated from each experimental image using the Photoshop color range tool and converted to grayscale. The contrast was then adjusted to 100% to set all dust pixels to a 255 grayscale value (*Figure 1—figure supplement 2B*). To visually display the average dust patterns, we used custom Matlab (MathWorks, Inc., Natick, Massachusetts) code (Groomogram) to generate average projections (*Figure 1—figure supplement 2C*, code included as *Source code 1*). Groomogram code works by averaging the grayscale pixel values (0 = no dust/255 = dust) for each pixel coordinate from all experimental images. The average pixel values for the entire image were converted to grayscale colormaps that represent the percent of flies with dust at a particular pixel coordinate (*Figure 1B*, *Figure 1—figure supplement 2D*).

*Calculating normalized dust pixels in a specified region*: pixels with 255 grayscale values (dust pixels), and located within the regions designated by standard masks were counted for each image (*Figure 1—figure supplement 3A,B*). The fractions of dust pixels per total number of pixels in each region as defined by the masks were calculated for each image sample. Normalization in *Figure 1D* was calculated as follows: fractions of dust pixels per sample were divided by the mean of the zero time point samples for each body part. Standard error was calculated from these values. Normalization in *Figure 3B* was calculated as follows: average dust pixel fractions per body region were divided by the maximum mean value for each body part. The ventral and dorsal abdominal data were combined for display in *Figure 3B* by calculating the mean of their normalized values.

## Behavioral recordings

*Camera setup*: an Edmund Optics 1312C color USB camera (Edmund Optics Corporation, Barrington, New Jersey) equipped with a Computar 25 mm f1.4 lens and an EX2C extender for C-mount (CBC AMERICA Corporation, Commack, New York) was used to record flies at 30 frames per second. *dTrpA1 behavioral analysis (undusted flies)*: this method was used for recording the flies shown in *Figure 2*. Flies were cold anesthetized at 1–2°C and placed into a 3 mm high and 13 mm diameter arena sitting directly on a teca solid state heat/cool cold plate (TECA Corporation, Chicago, Illinois). The flies were allowed at least 5 min to recover from the cold anesthesia and then recorded for 2 min when the cold plate temperature sensor read 21°C, 30°C, and then 21°C (post activation analysis) (*Figure 2—figure supplement 1*). *Dusted experiments:* Canton S or GAL4/*UAS-dTrpA1* flies were dusted individually in 24-well grooming chambers as described above. An acrylic plug was inserted into the hole on the adapter end and the aluminum slider was removed such that the fly could be recorded (*Figure 1—figure supplement 1D*). Flies were also shaken without dust and recorded as a control for the behavioral effects of the shaking. *Dusted/cooled experiments:* flies were shaken with or without dust, allowed to groom for 25 min at 30°C, cooled to 18°C, and their behavior was recorded for 3 min. After grooming for 24 min at 30°C, the flies were transferred to a 24-well aluminum grooming block that was preheated to 30°C. Just before 25 min, an acrylic plug was placed over the chamber opening and the block was placed on a TECA cold plate that was set to 18°C to rapidly cool the flies and shut off the dTrpA1 channel. Cleaning movements were scored for 3 min starting when the plate touched the cold plate surface (*Figure 3C*, *Figure 3—figure supplement 3*).

## Movement scoring guidelines and definitions

Pre-recorded video was manually scored using Noldus Observer XT 7.0 software (Noldus Information Technology, Wageningen, Netherlands). A list of cleaning movements was compiled and modified from

two different studies (*Szebenyi, 1969*; *Dawkins and Dawkins, 1976*). Flies clean their bodies with their front or hind legs as mutually exclusive events (*Figure 1A*). Front leg cleaning movements are directed to the head whereas the hind legs clean the abdomen, wings, and thoraces. Starts of cleaning events are defined when the legs are farthest anterior (hind leg movements) or posterior (front leg movements) of the body part before they sweep in the opposite direction. Transitions between movements were included at the end of the preceding movement. Wing cleaning with the hind legs and leg rubbing using the second legs were sometimes asymmetrically performed in that one leg performed cleaning of a particular body part while the contralateral leg was stationary.

*Eye and head cleaning*: legs sweep across the eyes, tops, and bottoms of the head as it is rotated. Most sweeps across the eyes occur in the absence of head rotation. *Antennae*: legs are directed towards the antennal region, often while the head is tilted slightly forward. The legs sometimes grab the antennae and pull them away from the head. *Proboscis*: sweeps down the length of the extended proboscis. The amount of proboscis extension does not factor into the definition as long as it is partially extended or seen moving with the legs. *Ventral head*: legs sweep the ventral side of the head with no proboscis extension. The front legs almost always sweep the bottom of the head/proboscis in parallel. Note, proboscis and ventral head cleaning were binned with eye and head cleaning for simplicity. *Whole head*: all of the head cleaning behaviors described above are binned. The whole head designation was used because it was not possible to distinguish between eye, head, and antennal cleaning when flies were dusted.

*Abdominal cleaning*: legs sweep the ventral or dorsal surface of the abdomen. The hind legs sometimes clean the genitals. This was included as abdominal cleaning for simplicity. *Wing cleaning*: each leg sweeps the ipsilateral dorsal side of the wings simultaneously, or one at a time. Alternatively, both legs clean a single wing with the ipsilateral leg cleaning the dorsal side and the contralateral leg cleaning the ventral side. Sweeping or kicking the ventral part of the wing with one or both legs was also referred to as wing cleaning. In this case, as the legs sweep anterior to posterior, the wings lift away from the body. *Thoracic cleaning*: hind legs sweep the thorax from anterior to posterior with one or both legs.

*Front/hind leg rubbing*: distal parts of the legs are rubbed together. Behavior begins at the moment that the legs touch each other. If the legs stop motion but remain together, it is defined as leg rubbing as long as they do not touch the ground. The middle legs were sometimes rubbed together with the front or the hind legs. This was included with front leg or hind leg rubbing depending on the leg pair being used with the second legs.

*Standing*: fly does not move more than a body length from the original scored position and does not perform any grooming movements. The beginning of standing is scored as soon as the last leg touches the ground. In some cases, the legs did not touch the ground to initiate the behavior and instead was scored at the point where the previous behavior was judged to have ceased. *Walking*: fly steps more than a body length with no pauses in leg movement. Walking is also defined as moving from one of the following locations in the chamber to another (floor, ceiling, or side). The beginning of walking is scored at the moment that one of the front legs initiates a step.

## Analysis of behavioral data

Manually scored behavioral video was analyzed as follows. *Bout frequency:* total number of starts of each behavior per fly. *Bout frequency per interval:* bout starts that fell within a designated time interval. *Marginal probability:* the average fraction of total bouts of a particular behavior. *Change from control:* the experimental parameter (either bout frequency or fraction of time) was subtracted from the control. *Statistical test:* control and experimental flies were compared using a Kruskal–Wallis test. Post-hoc Mann–Whitney U pairwise tests were then performed with Bonferroni correction. *Transition probability:* calculated by counting the number of transitions from each behavior to each other behavior. Next, we calculated the fraction of times that behavior i transitioned to behavior j given the total of number of transitions to all behaviors. Therefore, the probability of going from behavior i to all other behaviors sums to 1.

## Screening and categorizing grooming GAL4 lines

GAL4 lines used in this study were obtained from a screen designed to identify and manipulate neural circuitry involved in grooming behavior. We used a collection of enhancer-driven GAL4 lines to

genetically target different subsets of neurons with the neural activator *UAS-dTrpA1* (*Hamada et al., 2008*; *Pfeiffer et al., 2008*; *Jenett et al., 2012*). Over 1500 GAL4 lines expressing dTrpA1 were manually examined at 30–32°C for increased grooming compared with wild type. We manually scored the movements of 12 identified GAL4 lines that were sufficient to activate strong cleaning movements (*Figure 2*). GAL4 lines used in this study were sufficient to activate cleaning movements but not necessary for cleaning dust off their respective body parts. This was revealed by inhibition and dusting experiments using both *UAS-Shibire^{ts1}* or *UAS-TNT* (data not shown). GAL4 lines expressing GFP were stained and their patterns imaged using previously published methods (*Pfeiffer et al., 2010*).

## Computational model of hierarchical suppression among competing cleaning modules

Two neural network architectures were used to simulate the hierarchical suppression among competing cleaning modules: a *sensory gain model* (SGM, *Figure 4C*) and a *unidirectional inhibition model* (UIM, *Figure 4D*). Both neural networks were implemented using Matlab 7 (code included as *Source code 2*). Both networks are based on three layers: sensory layer, hierarchical layer, and winner-take-all layer.

*Sensory and hierarchical layers*: for both network implementations, we defined a vector of *activation levels*, **a**, to represent N neural modules (this is the hierarchical layer). For the UIM architecture, the activation levels are initially set to equal the vector of *amounts of dust*, **d** (this is the sensory input layer). The initial amount of dust was set to $0.9 + r$, where $r$ is a random value between 0 and 0.1, sampled from uniform distribution. This noise term is only a factor in *Figure 4B* (no hierarchy set among the modules) where slight differences in the relative dust levels determine which module is selected first. While for the SGM architecture, the vector $a = dw^s$, where $w^s$ is the vector of *sensory weights*. In UIM, $w^s$ is set to one (no sensory gain differences) and vector a is then multiplied by the $N \times N$ weight matrix, W, of which the diagonal = 1, and the upper triangle is populated by equal negative weights, $w = -0.5$. For SGM, $w^s$ is set to $w^s = [w_{max}, w_1, w_2, \ldots w_{min}]$, which insures that the sensory weights decrease from the first module to the last. Thus, for each iteration $i$, we update the activation levels as follows:

$$a^{i+1} = d^i w^s, \text{SGM}$$

$$a^{i+1} = a^i W, \text{UIM}$$

For simulating no hierarchical differences among the modules (shown *Figure 4B*), we set $w^s$ to 1, and the upper triangle of W to 0.

*Winner-take-all layer*: to simulate removal of dust from the body parts, this layer first finds the module with the highest activation level that was set by SGM or UIM. The position of the module with maximal activation level, *ma*, is determined: $ma = max(a)$. The winner-take-all layer is then simply a binary vector e containing all zeros and equals 1 only in the position *ma*, thus pointing to the body part where cleaning behavior will be executed. Next, a constant amount of dust, *dr*, is subtracted from the $ma^{th}$ position in vector d ($d_{ma}$). Thus, for each iteration $i$:

$$ma^i = max(a^i)$$

$$d^{i+1}_{ma} = d^i_{ma} - dr$$

Note: we did not explicitly simulate the circuitry required for implementation of the winner-take-all layer. Instead, we selected the module with maximal activation level (*ma*) by simply using the Matlab *max()* function. Winner-take-all neural networks can be modeled using all-to-all inhibitory connections between the different modules (*Houghton and Hartley, 1995*). Such explicit simulations of winner-take-all function might capture the empirically observed behavior even better but would require more complex modeling, going beyond the scope of the claims presented in this work.

*Constitutive activation of cleaning modules:* to implement constitutive cleaning of a particular body part, we held the dust level of a given cleaning module constant (at maximum dust level) (*Figure 4E,F*). This models the situation where the fly is receiving constant sensory input even in the absence of stimulus.

*Leg rubbing*: to implement leg rubbing, we assumed two pairs of legs. Both pairs were assigned an initial activation level (al; a 2 dimensional vector), which was directly proportional to the amount of dust on legs (dl). For each iteration $i$, we added a small, constant amount of dust to the pair of legs assigned to clean the body part with the maximal activation level ($ma$). If $ma =< 3$ ('anterior body parts') the first leg pair was assigned and if $ma > 3$ ('posterior body parts') the second leg pair was assigned to collect dust. Therefore, with each grooming iteration, the activation level of the associated pair of legs increases.

Leg rubbing (removal of dust from legs themselves) is initiated under two conditions: (1) the activation level of the pair crosses a stationary threshold (same for both pairs) and (2) the activation level of the pair is above the activation level of any body part (above $ma$), including the other pair of legs. The latter condition assures that leg rubbing is still part of the competitive hierarchy of cleaning modules (*Figure 5C,D*).

*Model output and display*: the output of the model is the position of the cleaning module with the maximum activation level at iteration $i(ma^i)$. Thus the output is an array with rows corresponding to $ma^i$ (active cleaning module) and columns corresponding to iteration $i$ ('time'). This array was used to generate the ethograms shown in *Figure 4* and *Figure 5*.

## Acknowledgements

We thank: Gerald Rubin and Lab for access to the GAL4 collection; James Truman for suggestions about quantification of dust patterns; Eric Hoopfer and Kristin Branson for providing Matlab analysis code, discussions on the work, and feedback on the manuscript; Vivek Jayaraman, Roland Strauss, Ulrike Heberlein, Scott Sternson, Adam Hantman, Shaul Druckmann, Stephanie Albin, Claire McKellar, Jon-Michael Knapp, and Carmen Robinett for feedback on the project and manuscript; Janelia Mechanical Engineering and Fabrication group for the grooming chambers; Janelia Fly Core for providing fly stocks; Rachel Wilson for initial guidance on the project; Daniel Bullock for critical discussions; and Stephen Huston for key ideas on parallel models. This work is dedicated to the memory of Frank M Midgley, Jr. (1971–2014), colleague, friend, and partner.

## Additional information

### Funding

| Funder | Author |
| --- | --- |
| Howard Hughes Medical Institute | Julie H Simpson |

The funder had no role in study design, data collection and interpretation, or the decision to submit the work for publication.

### Author contributions

AMS, Conception and design, Acquisition of data, Analysis and interpretation of data, Drafting or revising the article; PR, BDM, Developed the computational model, Drafting or revising the article; PC, Acquisition of data, Drafting or revising the article; SH, Drafting or revising the article, Contributed unpublished essential data or reagents; FMM, Wrote custom software to analyze grooming data, Drafting or revising the article; JHS, Conception and design, Drafting or revising the article

## Additional files

### Supplementary files

• Supplementary file 1. This file contains design plans for the grooming chambers used in this study (shown in *Figure 1—figure supplement 1*).

• Source code 1. This Matlab code was used to display the average projections of dust patterns on the different body parts (shown in *Figure 1—figure supplement 2C*). Groomogram code works by averaging the grayscale values from multiple images of the same size for each pixel coordinate.

• Source code 2. This code for Matlab will simulate dust-induced fly grooming behavior. Simulations are shown in *Figure 4* and *Figure 5C,D*.

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
