## [Decision Letter]

Thank you for sending your work entitled “A suppression hierarchy among competing motor programs drives sequential grooming in *Drosophila*” for consideration at *eLife*. Your article has been favorably evaluated by K (Vijay) VijayRaghavan (Senior editor) and 3 reviewers, one of whom, Ronald L Calabrese, is a member of our Board of Reviewing Editors.

The Reviewing editor and the other reviewers discussed their comments before we reached this decision, and the Reviewing editor has assembled the following comments to help you prepare a revised submission.

The authors describe experiments on fruit fly grooming when the entire body is dusted. This paradigm sets up a distinctive motor sequence in which grooming proceeds and allowed the testing of the organizational principles governing how dusted flies groom their bodies with sequential movements. Using genetically targeted activation of neural subsets, the authors drove different components of the grooming sequence that clean individual body parts. They then performed competition experiments to show that the different motor components are organized into a suppression hierarchy so that components that occur first suppress those that occur later. They then make a simple 'neural' model with two layers, a hierarchical layer of the independent cleaning movements and a winner take all layer, which nicely captures their data. Cleaning one body part reduces the sensory drive to its cleaning movement, which relieves suppression of the next movement, allowing the grooming sequence to progress down the hierarchy. They then discuss the implications of their finding for how sequential behaviors are performed and how simple behavioral choices are made.

The paper is beautifully illustrated and all necessary data are presented in the figures and figure supplements. The writing is concise and in general clear, and the Discussion is illuminating, but could be expanded a bit. The paper is highly interesting and innovative and provides a mechanistic explanation for how motor sequences can be organized. It should be of wide interest to the neuroscience community.

Both the outside reviewers and the Reviewing editor found this paper a pleasure to review and the authors are to be commended for an exceptional piece of work beautifully presented. We encourage the authors to integrate some of the supplemental data into the text, particularly Figure 1—figure supplement 3 and Figure 1—figure supplement 4 and Figure 3—figure supplement 1 and Figure 3—figure supplement 2.

The outside reviewers had several concerns that are intended to make this interesting paper yet more interesting. The reviews are very complementary and require only some rewriting to add details, some more discussions of implications and limitations, and limited amount of further analysis. We anticipate that the authors should be able to quickly answer these concerns, despite their considerable number and resubmit this exciting work. No new experiments are requested.

Specific comments:

Reviewer #1:

The paper is very cleanly written.

I encourage the authors to integrate some of the supplemental data into the text, particularly Figure 1—figure supplement 3 and Figure 1—figure supplement 4 and Figure 3—figure supplement 1 and Figure 3—figure supplement 2.

Methods are a little terse; particularly the models and their outputs could be explained more fully.

Reviewer #2:

This is an altogether impressive piece of work. The approach, although incredibly complex in its details, is extremely clear-headed in how it is organized and presented. The results are spectacular and, at least to me, completely unexpected. They screened more than 1500 Drosophila GAL4 stocks created at Janelia that expressed a temperature-sensitive cation channel (dTrpA1) in all the GAL4-expressing neurons. From these, they found 12 lines that reliably affected different aspects of grooming behavior. They then devised behavioral and anatomical tests to assay the sequence of behaviors that normally occurs in normal grooming. They then used activation of the components of grooming (using the legs to wipe dust from different body parts) to test whether the sequencing of the grooming components (which they call “modules”) depends upon a series or a parallel mechanism. (They come down strongly in favor of a parallel mechanism.) Both the behavioral and genetic tools they devised are extremely clever (horrendously tedious, but conceptually clear), and the analyses are appropriate and convincing. The major conclusion (that there are separable modules of grooming that inhibit one another in a hierarchical order) is well documented and, for the most part, clearly presented.

There are, despite these great strengths, a few problems with the study. Some of these problems are mentioned briefly, but others are not. The major ones are:

1) We are told nothing about the behavioral phenotypes of the 12 lines at the elevated temperature. Ideally, it would be nice to know that the obsessive activation of some aspect of grooming is the only behavioral anomaly that they show.

2) From the extreme variety of neurons that express GAL4 in the different Drosophila lines (there are vast differences in both total cell numbers and anatomical distributions), it is remarkable that these lines express such specific behavioral activations when heated. I was disappointed that there was no discussion of this remarkable observation. I could think of several alternatives, but was disappointed that people who are much more familiarity with these animals did not address this question, at least briefly.

3) As far as they carried it, the modeling study was quite useful. Having a fairly simple model that matches the data so well, even some data that they didn't use to construct the model is very comforting. They made a big point, however, that there were two possible explanations (a serial one and a parallel one) for their results, but they attempted to model only the parallel one. Showing that any serial model has serious problems would have rounded off the argument in a satisfying way. They discuss why the serial option would probably not work, but I found this Discussion unconvincing.

4) I felt blind-sided when, in the Results, I was told that “the frequent alternations between cleaning the body parts and leg rubbing … are an important and prominent feature in normal grooming behavior”. If these movements are so important, why are they mentioned only as a second-order test of the model and not in any of the studies of the behaviors? Minimally, at the end of the first paragraph in Results the authors could tell us that there is another feature of grooming that will be detailed later. In fact, the initial transitions diagram (Figure 1) has a hint of this alternation; it could be mentioned in discussing this diagram.

5) They mention that silencing the same GAL4-expressing neurons does not eliminate the behaviors that activating them elicits. Again, I was looking forward to their ideas about why these neurons are sufficient but not necessary, but they did not go there.

Reviewer #3:

Understanding how specific sequences of behavior are executed remains challenging, with a number of alternative models fitting data in different contexts. In this work, Seeds et al. describe a new behavioral sequence that underlies grooming behavior, and take advantage of quantitative analyses and genetic manipulations that allow them to get at the organization of this behavior. Intriguingly, these studies support a parallel track model in which sensory input, combined with different sensory gains and a winner take all competition recapitulate many of the core features of fly behavior. As this system is the first to capture a sequence in which the sensory input that drives the behavior is the same across modules, it represents a powerful paradigm for the dissection of its neural control. This work is quantitative, rigorous, makes a general point and is appropriate for *eLife* with some modest changes I describe below.

My only concern regarding the fly behavioral data is that I believe that Figure 1 is oversimplified, and hence misleading. The time traces in Figure 1 and the fraction changing in Figure 1 demonstrate that the transition from one behavior mode to another is quite smoothly graded, a feature that is eliminated from the very simple view in Figure 1. I would therefore request that the authors extend the type of display in Figure 1, developing a time-dependent transition probability model. A simple way to do this would be to cut the ethogram data into 5 minute segments, and replot the transition probability model (thresholded, as the authors do already). This kind of representation should give a more accurate sense of how the behavior changes progressively.

My second concern has to do with the explanation of the structure of the models the authors use. As far as I can tell, the model structure using sensory feedback and the winner take all layer should be deterministic – at whatever level of sensory gain is initiated, combined with the present level of dust, only one module should be selected (and should persist until the level of dust on that body part drops sufficiently for another module to win). However, the data in Figure 4 show that, like real flies, at most time points, the models have mixed character, with flies alternating between different activities (which change progressively). I understand how leg grooming is exceptional, in the sense that its sensory drive increases periodically (giving an oscillatory character to the behavior), but for all other grooming behaviors, the sensory drive can only decline. Is there a noise term that is not described in the text? If so, what is its form? If not, can the authors provide a more accessible description for why their models are not deterministic in time?

---

## [Author Response]

Reviewer #1:

*The paper is very cleanly written*.

*I encourage the authors to integrate some of the supplemental data into the text, particularly*
Figure 1—figure supplement 3 and Figure 1—figure supplement 4
*and*
Figure 3—figure supplement 1 and Figure 3—figure supplement 2

We intended for the main figures to provide a concise view of the results, while the supplemental material expands on the complexity of the data. We made the following additions to the text and figure legends to insure the reader receives full benefit of the figure supplements. Particularly we have tried to make it easier to logically link to figure supplements from the Results section and the primary figure legends. The ease of viewing those figure supplements that complement the primary figures in *eLife*’s online format should facilitate this.

Figure 1—figure supplement 3: We added text to highlight the relationship between this figure supplement and Figure 1. We now mention in the Figure 1 legend that the data used to generate the panel can be found in Figure 1—figure supplement 3: “Figure panel is compiled from data shown in Figure 1—figure supplement 3.” We also mentioned in the supplementary Figure legend: “Data shown here is compiled and plotted in Figure 1.” This should allow the reader to more easily see the relationship between the two.

Figure 1—figure supplement 4: We now refer to specific panels of this figure supplement in the first section of the Results. Text now reads: “In response to dust on their bodies, flies groomed with a predictable progression of targeted cleaning movements (Figure 1, Figure 1—figure supplement 3, Figure 1—figure supplement 4). This progression required dust, as undusted flies did not groom in a sequential progression (Figure 1—figure supplement 4).” We highlight Figure 1—figure supplement 4 more in another sentence of the first Results section that now reads: “That is, the progression was not absolutely unidirectional as flies returned to cleaning earlier body parts even though they had already started cleaning later body parts (referred to as return cleaning, discussed more below) (Figure 1 and Figure 1—figure supplement 4, example ethograms).”

Figure 3—figure supplement 1: We now highlight the visual match between the grid in Figure 3 and the average dust distributions on the fly body parts shown in Figure 3—figure supplement 1. We added text to the Results section, main Figure legend, and Figure supplement legends to make this point. The Results text now emphasizes the relationship between the two:

“Figure 3 illustrates the outcome of these experiments, which matched the prediction of the hierarchical suppression hypothesis (compare Figure 3—figure supplement 1 to Figure 3).”

The legend describing Figure 3 now states: “Data used to generate the grid is shown in Figure 3—figure supplement 1.” The legend describing Figure 3—figure supplement 1 now states:

“Data shown here is compiled and plotted in Figure 3.”

The Results describing Figure 3 now more specifically refers to Figure 3—figure supplement 1: “GAL4 lines that activated abdominal (R24B03), wing (R53A06) or all posterior cleaning modules (R45G01) showed little impairment in removing dust from their heads (Figure 3, Figure 3—figure supplement 1, see R24B03, R53A06, and R45G01).” Figure 3—figure supplement 2: In the Results section, the Figure supplement is more specifically referred to highlight each example. Specific example 1: “For example, when a fly carrying a GAL4 driver that activates abdominal cleaning (R24B03) was stimulated in the presence of whole-body dust, it first cleaned the head, then abdomen (just as wild-type flies do), but then persisted in cleaning its abdomen instead of proceeding to its wings and thorax (Figure 3, Figure 3—figure supplement 2, see R24B03).” Specific example 2: “Like wild-type flies, these GAL4 lines cleaned their heads at the onset of dusting, indicating that the normal progression of cleaning was preserved up to the point where it was blocked (Figure 3—figure supplement 2, see R24B03, R53A06, and R45G01).”

*Methods are a little terse; particularly the models and their outputs could be explained more fully*.

We have changed and added text in the methods sections to improve clarity. Sections modified include: Grooming apparatus; Dusting flies for body surface imaging or video recordings; Quantification of body surface dust patterns; Behavioral recordings; Movement scoring guidelines and definitions; Computational model of hierarchical suppression among competing cleaning movements. The Methods pertaining to the model were expanded to describe the outputs and to provide greater conceptual linkage to the Results presented in Figure 4 and Figure 5.

Reviewer #2:

*1) We are told nothing about the behavioral phenotypes of the 12 lines at the elevated temperature. Ideally, it would be nice to know that the obsessive activation of some aspect of grooming is the only behavioral anomaly that they show*.

We deliberately chose lines for this study that were extremely specific for grooming behaviors. These lines were assessed for gross behavioral phenotypes and motor coordination. This was done both in the course of their selection while screening the GAL4 collection and in doing the experiments in this paper. These lines constitutively clean particular body parts and show no other obvious behavior. One line shows a slight motor impairment when it is walking (R23A07, Figure 2–figure supplementary video 1). We emphasize that the readers can watch the representative videos of different GAL4/dTrpA1 lines to see what the phenotypes look like. We added two additional videos (now a total of eight) so all of the lines used in the experiments in Figure 3 and Figure 5 are shown. The specific cleaning phenotypes for each line are described in the video Figure legends and now also describe any other phenotypes that are not conveyed by the Figure 2 ethograms or grid. Most video legends say: “No other overt phenotypes were observed.” The R23A07 legend now says: “Displayed minor walking defect that was unrelated to the cleaning phenotype.”

We also reference these videos in the Results section of Figure 2: “We identified different GAL4 lines that caused exclusive cleaning of each body part, including the head, abdomen, wings, and legs (Figure 2, Figure 2—figure supplement 1, Figure 2—figure supplement 2, see Figure 2–figure supplementary videos 1–8 for representative videos and more detail regarding the behavioral phenotypes).” The ethograms in Figure 2 provide a nice way to get an idea of the phenotypes of these lines. We added description to the Figure 2 legend so that the readers get a fuller understanding of the ethograms for each line: “White space in each ethogram represents time spent walking or standing in place.” We also now address this issue in discussing our rational for selecting lines used in the hierarchy experiment in Figure 3: “For this experiment, we selected lines that could induce a particular cleaning module and lacked additional phenotypes that might confound our grooming-specific interpretations (Figure 2, lines labeled in red, Figure 2–figure supplementary videos 1–8).”

*2) From the extreme variety of neurons that express GAL4 in the different Drosophila lines (there are vast differences in both total cell numbers and anatomical distributions), it is remarkable that these lines express such specific behavioral activations when heated. I was disappointed that there was no discussion of this remarkable observation. I could think of several alternatives, but was disappointed that people who are much more familiarity with these animals did not address this question, at least briefly*.

We now address the expression patterns targeted by the GAL4 lines in the last Discussion section that also covers comment 5 from this reviewer (last three paragraphs of the discussion). In brief, we make two points about these expression patterns as they relate to the behavioral activation phenotypes: 1) Only a fraction of the cells in each expression pattern are responsible for the phenotypes as GAL4 lines do not necessarily target functionally related neurons. 2) The cells targeted in each pattern may be related in function (e.g. such as in the *Drosophila fruitless* circuit). We also described what the actual cell types might be in this discussion section. Namely we discuss two possible neural types as capable of activating such specific behavior: 1) sensory neurons 2) Command-like/decision neurons.

*3) As far as they carried it, the modeling study was quite useful. Having a fairly simple model that matches the data so well, even some data that they didn't use to construct the model is very comforting. They made a big point, however, that there were two possible explanations (a serial one and a parallel one) for their results, but they attempted to model only the parallel one. Showing that any serial model has serious problems would have rounded off the argument in a satisfying way. They discuss why the serial option would probably not work, but I found this Discussion unconvincing*.

Two major aspects of our data led us to reject the serial model and focus on computationally simulating the parallel model. First, the cleaning modules have parallel sensory inputs and stimulation of one input does not result in a complete sequence of movements, as might be predicted by an activation chain. Second, a serial model does not predict the return cleaning among the modules that occurs in wild type flies (return cleaning is described more in the next paragraph). We reasoned that an activation chain model would simply result in a deterministic sequence of simulated grooming movements, rather than this return cleaning. This said we make no claim that some degree of internal triggering of cleaning modules (through an activation chain) does not exist. In other words, a ‘mixed model’ could explain the empirically observed grooming behavior even better. In this case, execution of a cleaning module is selected by two components: winner-take-all (parallel model) and internal triggering from the previous cleaning module (serial model). All we claim in this work is that the parallel model alone is fully sufficient to reproduce the major aspects of grooming sequence discussed in this manuscript. We revised the Results and Discussion to better set up and then state our argument against an activation chain as the primary mediator of the grooming sequence. The manuscript has been changed to address this and other related comments from reviewers two and three. We now focus more attention on the return cleaning that occurs among the cleaning modules. We do this by first establishing that the flies clean a body part, then clean the next, but still return to clean previous body parts. These returns by the cleaning modules in wild type flies are now emphasized more in the Results that describe Figure 1. Next, a new paragraph was added on the return cleaning in our description of the output of the model (in reference to Figure 4). Finally, we argue in the Discussion that return cleaning is not what is expected of a simple activation chain model, but can be explained by our parallel model.

*4) I felt blind-sided when, in the Results, I was told that “the frequent alternations between cleaning the body parts and leg rubbing … are an important and prominent feature in normal grooming behavior”. If these movements are so important, why are they mentioned only as a second-order test of the model and not in any of the studies of the behaviors? Minimally, at the end of the first paragraph in Results the authors could tell us that there is another feature of grooming that will be detailed later. In fact, the initial transitions diagram (*Figure 1*) has a hint of this alternation; it could be mentioned in discussing this diagram*.

We apologize that this was not clear and now emphasize leg rubbing more in the first paragraph of the Results section. There was already text in this paragraph that referred to leg rubbing in grooming behavior. In fact, our primary purpose for including Figure 1 was to show the prominent transitions between body sweeps and leg rubbing (see below for additional discussion of Figure 1). We have revised the section to further emphasize the point (including a reference to watch the supplementary video) and now use a term to describe this feature of grooming (cyclic transitions). The text now reads: “As the flies progressed through the grooming sequence, each bout of body part cleaning featured cyclic transitions between sweeps of the targeted region and rubbing of the legs against each other (Figure 1, Figure 1—figure supplement 5, Video 1). These cyclic bouts of leg rubbing likely occurred when the legs accumulated sufficient dust from sweeping the body, which was then rubbed towards the distal leg parts and removed.”

To make this point even more clear, we added additional text that mentions these cyclic transitions in the second to last sentence of the first Results paragraph: “Thus, flies groom by gradually cleaning different body parts in bouts that are characterized by cyclic transitions between body cleaning sweeps and leg rubbing (shown as a series of transition diagrams in Figure 1—figure supplement 6).”

*5) They mention that silencing the same GAL4-expressing neurons does not eliminate the behaviors that activating them elicits. Again, I was looking forward to their ideas about why these neurons are sufficient but not necessary, but they did not go there*.

We now address this point in a new (last) section of the Discussion that also covers point two from this reviewer: “Neurons that activate cleaning modules”. In brief, we make the point that the neurons may be sensory or command-like neurons. We describe that such command-like neurons do not necessarily follow a strict definition as command neurons in being both necessary and sufficient. Alternatively (not described in the discussion), redundancy exists among the grooming sufficient neurons, and each GAL4 line only targets a fraction of the total population sufficient to drive a specific cleaning movement. This could obscure results of an experiment to test neural necessity in grooming because the entire population would not be inhibited through expression of neural inhibitors.

Reviewer #3:

*My only concern regarding the fly behavioral data is that I believe that*
Figure 1
*is oversimplified, and hence misleading. The time traces in*
Figure 1
*and the fraction changing in*
Figure 1
*demonstrate that the transition from one behavior mode to another is quite smoothly graded, a feature that is eliminated from the very simple view in*
Figure 1*. I would therefore request that the authors extend the type of display in*
Figure 1*, developing a time-dependent transition probability model. A simple way to do this would be to cut the ethogram data into 5 minute segments, and replot the transition probability model (thresholded, as the authors do already). This kind of representation should give a more accurate sense of how the behavior changes progressively*.

We agree with reviewer three that Figure 1 is oversimplified and misleading as a representation of what serial grooming actually looks like. To avoid further confusion by other readers on this point we removed Figure 1 and now simply emphasize the priority order of grooming in a new the last sentence of the first Results section in the text: “The priority order for cleaning the different body parts is: eyes > antennae > abdomen > wings > thorax.”

Reviewer 3 also suggested that we produce time-binned diagrams (like Figure 1) from the twenty-five minute annotated video of dusted flies. Figure 1 was not intended to be used as evidence supporting the grooming sequence, but to describe two other features of dust induced grooming behavior. First, it shows the frequent transitions between body cleaning and leg rubbing. Second, it shows that the progression of grooming is discontinuous, in that walking frequently interrupts cleaning. Descriptions of this are in the first Results section, and we have added text to make this point more clear (see comments on this in relation a comment from reviewer two). For these reasons, we respectfully ask to keep Figure 1 in its current place. However, we generated the five-segmented diagrams requested by the reviewer and propose to include it as a figure supplement (Figure 1—figure supplement 6) to add more description of the progression of grooming. Figure 1—figure supplement 6 is used to further support the conclusion of the first Results section. The second to last sentence of this section now reads: “Thus, flies groom by gradually cleaning different body parts in bouts that are characterized by cyclic transitions between body cleaning sweeps and leg rubbing (shown as a series of transition diagrams in Figure 1—figure supplement 6).” This sentence also addresses an issue from reviewer two that there was insufficient description of the body cleaning and leg rubbing bouts.

*My second concern has to do with the explanation of the structure of the models the authors use. As far as I can tell, the model structure using sensory feedback and the winner take all layer should be deterministic – at whatever level of sensory gain is initiated, combined with the present level of dust, only one module should be selected (and should persist until the level of dust on that body part drops sufficiently for another module to win). However, the data in*
Figure 4
*show that, like real flies, at most time points, the models have mixed character, with flies alternating between different activities (which change progressively). I understand how leg grooming is exceptional, in the sense that its sensory drive increases periodically (giving an oscillatory character to the behavior), but for all other grooming behaviors, the sensory drive can only decline. Is there a noise term that is not described in the text? If so, what is its form? If not, can the authors provide a more accessible description for why their models are not deterministic in time?*

We do use a noise term in our model that varies the initial amount of dust on the cleaning modules in the upper ten percent of the total dust amount (see revised Methods). This causes variability in which modules are selected first when there is no suppression hierarchy (i.e. as in Figure 4). This is not, however, what causes the alternations among the cleaning movements described by the reviewer. Alternations occur as the activation levels between the movements get closer and closer through cleaning. For example, module 1 occurs first in the sequence and proceeds until the dust level on its corresponding body part is reduced enough that the next dirty module 2 now has a higher relative activation level. Module 1 is not yet clean, but clean enough that its sensory drive is less than module 2. Once module 2 cleans its body part, thus reducing its own sensory drive, the activation level of module 1 is now above 2. These two modules will alternate back and forth until their activation levels fall below module 3. Now these alternations will occur among all three modules. Therefore, the sequence has a sequential nature but, at the point where the activity levels become more similar through cleaning, the behavior takes on this alternating character (we now refer to this as return cleaning in the paper – i.e. returning to clean previously cleaned parts). This is apparent in the ethograms of actual flies after they are dusted, and in the model (Figure 1, Figure 4, Figure 5). Although the appearance of the different movements in a particular order (Figure 1) may be thought of as deterministic, the return cleaning between the cleaning movements is more probabilistic. We added a paragraph to the Figure 4 Results section to describe this property of the model.